# Approach to Studies on Podocyte Lesions Mediated by Hyperglycemia: A Systematic Review

**DOI:** 10.3390/ijms26188990

**Published:** 2025-09-15

**Authors:** Jordana Souza Silva, Camila Botelho Miguel, Alberto Gabriel Borges Felipe, Ana Luisa Monteiro dos Santos Martins, Renata Botelho Miguel, Maraiza Oliveira Carrijo, Laise Mazurek, Liliane Silvano Araújo, Crislaine Aparecida da Silva, Aristóteles Góes-Neto, Carlo José Freire Oliveira, Juliana Reis Machado, Marlene Antônia Reis, Wellington Francisco Rodrigues

**Affiliations:** 1Postgraduate Course in Health Sciences, Federal University of Triangulo Mineiro, Uberaba 38025-180, MG, Brazil; jordana.ss13@hotmail.com (J.S.S.); albertogabrielborges@gmail.com (A.G.B.F.); ana.uisamello@gmail.com (A.L.M.d.S.M.); lili_silvano@yahoo.com.br (L.S.A.); crislaine.silva@uftm.edu.br (C.A.d.S.); carlo.oliveira@uftm.edu.br (C.J.F.O.); juliana.machado@uftm.edu.br (J.R.M.); marlene.reis@uftm.edu.br (M.A.R.); 2Multidisciplinary Laboratory of Scientific Evidence, Department of Biosciences, University Center of Mineiros, Mineiros 75833-130, GO, Brazil; camilabotelho@unifimes.edu.br (C.B.M.); maraiza.carrijo@academico.unifimes.edu.br (M.O.C.); laisemazurek@hotmail.com (L.M.); 3Genetics Laboratory, Institute of Biotechnology, Federal University of Uberlandia, Uberlândia 38402-018, MG, Brazil; renatabotelhonutri@gmail.com; 4Molecular and Computational Biology of Fungi Laboratory, Department of Microbiology, Instituto de Ciências Biológicas, Universidade Federal de Minas Gerais, Belo Horizonte 31270-901, MG, Brazil; arigoesneto@gmail.com

**Keywords:** podocytes, diabetic neuropathy, physiopathology

## Abstract

Podocyte injury is a central event in the pathogenesis of diabetic nephropathy (DN). We conducted a systematic review across four major databases, identifying 7769 records and including 130 studies that met predefined eligibility criteria. Methodological quality was assessed with Joanna Briggs Institute tools, yielding a mean score of 81.3%, indicating overall moderate-to-high rigor despite design-contingent limitations. Publication activity was sparse until 2018 but increased markedly thereafter, with more than 80% of studies published between 2019 and 2025. Temporal analyses confirmed a strong positive trend (*p* = 0.86, *p* < 0.0001), reflecting the rapid expansion of this field. Study designs evolved from early human-only descriptions to integrated multi-model approaches combining human tissue, animal experiments, and in vitro systems, thus balancing clinical relevance with mechanistic exploration. Geographically, Asia emerged as the leading contributor, complemented by increasing multinational collaborations. Mechanistic synthesis highlighted five reproducible pillars of podocyte injury: slit-diaphragm and adhesion failure, mTOR–autophagy–ER stress disequilibrium, mitochondrial and lipid-driven oxidative injury, immune, complement, and inflammasome activation, and epigenetic and transcriptomic reprogramming. Collectively, these findings underscore a convergent mechanistic cascade driving podocyte dysfunction, while also providing a framework for therapeutic interventions aimed at restoring barrier integrity, metabolic balance, and immune regulation in DN.

## 1. Introduction

Podocytes are highly specialized epithelial cells that cover the external surface of glomerular capillaries along the glomerular basement membrane (GBM), forming the outer layer of the glomerular filtration barrier (GFB) and preserving its integrity through a complex cytoskeleton, intercellular junctions, and interdigitating foot processes [1]. Under pathological conditions such as diabetic nephropathy (DN), terminally differentiated podocytes may undergo epithelial–mesenchymal transition (EMT), a reprogramming linked to slit-diaphragm instability, loss of adhesion, cytoskeletal disorganization, and barrier failure [2,3]. Disruption of these structural and molecular elements impairs glomerular function and promotes proteinuria and renal disease progression, underscoring the central role of podocytes in GFB homeostasis [1].

Diabetes mellitus (DM) is expanding globally, and diabetic kidney disease (DKD/DN) remains one of its most common and costly complications, with prevalence estimates of ~30–50% among people with type 2 diabetes across studied populations and a substantial impact on morbidity, mortality, and health-care expenditure [4,5,6]. Clinically, older age, albuminuria, and reduced estimated glomerular filtration rate (eGFR) mark a trajectory from early podocyte stress to kidney failure, reinforcing the need for timely detection and intervention [5,6].

Histopathologically, DN features GBM thickening, mesangial expansion, and progressive glomerulosclerosis, with Kimmelstiel–Wilson nodules in advanced stages and tubulointerstitial injury comprising tubular atrophy, interstitial fibrosis, and arteriolosclerosis, all converging on loss of renal function [7,8]. Human and experimental evidence indicates that podocyte injury, foot-process effacement, slit-diaphragm disruption, and progressive depletion, emerges early and predicts the transition to albuminuria and eGFR decline, positioning the podocyte as both a mechanistic hub and a therapeutic target [1,9].

At the mechanistic level, chronic hyperglycemia activates interlocking metabolic, hemodynamic, and inflammatory pathways, including the polyol and hexosamine routes, protein kinase C, the renin–angiotensin–aldosterone system, and advanced glycation end products, that culminate in oxidative stress, mitochondrial dysfunction, cytoskeletal remodeling, and slit-diaphragm failure [10,11,12]. In podocytes, recurrent molecular targets include mTOR signaling, autophagy and endoplasmic-reticulum stress, and transcriptional axes such as TXNIP–mTOR and Wnt/β-catenin, alongside alterations of structural and adhesion proteins (nephrin/podocin, integrins), which together drive detachment, effacement, and apoptosis [3,13,14]. These pathways connect to clinical readouts through conventional biomarkers (albuminuria, eGFR) and, increasingly, through omics signatures that may refine risk stratification and support precision medicine [6,7,15].

Therapeutically, rigorous glycemic and blood-pressure control remains foundational, while sodium glucose cotransporter-2 inhibitors (SGLT2i) reduce albuminuria, modulate metabolic and hemodynamic stressors, and slow DN progression, with growing interest in combination strategies involving GLP-1 receptor agonists and mineralocorticoid receptor antagonists [6,16,17]. Nevertheless, the persistent global burden and interindividual heterogeneity highlight knowledge gaps in how systemic drivers integrate with intracellular hubs and organellar/structural targets within podocytes along the disease continuum [5,6].

Given the high prevalence and impact of DN in type 2 diabetes, the centrality of podocyte injury, and the multiplicity of implicated pathways, there is a need for an integrative synthesis that mechanistically organizes links between systemic drivers, signaling modules, and organellar/structural lesions in podocytes, and aligns these with histologic and clinical outcomes [5,6,12]. Accordingly, this study aims to systematize and integrate evidence on podocyte injury in DN, outlining a mechanistic framework that connects hyperglycemia and comorbid drivers to intracellular hubs and barrier/adhesion failure, with implications for biomarkers and therapeutic targeting in diabetic kidney disease [13,14,15,18].

Unlike previous reviews, this work uniquely integrates temporal, methodological, and geographical trends with mechanistic insights, providing a comprehensive framework to guide both experimental and translational research.

## 2. Results

We identified 7769 records across four databases (Cochrane Reviews, MEDLINE/PubMed, LILACS, and Embase). After removing 5502 duplicates, 2267 records were screened at title/abstract level, of which 2021 were excluded. We sought 246 full-text reports and were unable to retrieve 32; thus, 214 reports were assessed for eligibility. Of these, 84 were excluded for predefined reasons, no podocyte injury (n = 46), no diabetes mellitus (n = 3), or no mechanistic data (n = 35), yielding 130 studies included from databases. Searches via other sources (websites, organizations, and citation chasing) located 18 additional reports; 6 could not be retrieved and 12 were assessed in full. None met the inclusion criteria (most commonly due to lack of podocyte injury), so no further studies were added. The full selection pathway is presented in Figure 1.

Methodological quality and risk of bias were appraised using the Joanna Briggs Institute (JBI) Critical Appraisal Tools. Across the 130 included studies, the mean methodological quality was 81.25% (SD 18.84; CV 23.19%), with scores ranging from 31.25% to 100.00%. Five studies achieved a perfect 100%, whereas three scored below 50% (31.25–37.50). As expected for observational designs, the most frequently unmet items concerned the use of independent control groups, within-participant controls, and the explicit identification and management of confounding variables. Despite these design-contingent limitations, the overall evidence base exhibited moderate-to-high methodological quality and no indication of bias likely to overturn the principal findings (Table 1).

**Table 1 ijms-26-08990-t001:** Assessment of methodological quality and risk of bias of eligible studies (jbi).

Author	Year	Methodological Quality—%
Balint et al. [19]	2023	87.50
Zhou et al. [20]	2017	62.50
Yamashiro et al. [21]	2024	68.75
Ivanac-Janković et al. [22]	2015	62.50
Carson et al. [23]	2014	81.25
Arslan et al. [24]	2025	81.25
Shetty et al. [25]	2021	68.75
Ceol et al. [26]	2012	62.50
Canney et al. [27]	2020	81.25
Esselman et al. [28]	2025	75.00
Denhez et al. [29]	2020	81.25
Audzeyenka et al. [30]	2020	87.50
Hayashi et al. [31]	2020	75.00
Chen et al. [32]	2020	87.50
Angeletti et al. [33]	2020	87.50
Albrecht et al. [34]	2023	81.25
Endlich et al. [35]	2018	62.50
Fujimoto et al. [36]	2020	81.25
Hu et al. [37]	2019	81.25
Chen et al. [38]	2024	50.00
Fiorina et al. [39]	2014	87.50
Fang et al. [40]	2021	81.25
Hou et al. [41]	2020	87.50
Han et al. [42]	2024	87.50
Holderied et al. [43]	2015	62.50
Gujarati et al. [44]	2024	87.50
Cao et al. [45]	2021	87.50
Jiang et al. [46]	2020	81.25
Fu et al. [47]	2020	50.00
Hu et al. [48]	2019	87.50
Kimura et al. [49]	2008	81.25
Lei et al. [50]	2024	87.50
Hu et al. [51]	2023	87.50
Kondapi et al. [52]	2021	87.50
Hu et al. [53]	2024	87.50
Kondapi et al. [54]	2021	87.50
Shahzad et al. [55]	2022	87.50
Kawaguchi et al. [56]	2021	87.50
Jiang et al. [57]	2022	87.50
Inoki et al. [58]	2011	87.50
Gödel et al. [59]	2011	87.50
Langham et al. [60]	2002	93.75
Bai et al. [61]	2018	87.50
Wang et al. [62]	2020	87.50
Lai et al. [63]	2020	87.50
Hudkins et al. [64]	2022	87.50
Kostic et al. [65]	2020	75.00
Liebisch et al. [66]	2020	75.00
Liu et al. [67]	2022	87.50
Li et al. [68]	2025	87.50
Li et al. [69]	2025	87.50
Lu et al. [70]	2021	87.50
Liang et al. [71]	2020	93.75
Hu et al. [72]	2025	87.50
Lv et al. [73]	2025	87.50
Lu et al. [74]	2021	87.50
Miyauchi et al. [75]	2009	87.50
Martins et al. [76]	2023	87.50
Lizotte et al. [77]	2023	87.50
Lee et al. [78]	2018	81.25
Löwen et al. [79]	2021	87.50
Lu et al. [80]	2023	87.50
Li et al. [81]	2024	87.50
Nishad et al. [82]	2021	87.50
Petrica et al. [83]	2021	100.00
Matoba et al. [84]	2021	87.50
Palmer et al. [85]	2021	100.00
Naito et al. [86]	2023	87.50
Pan et al. [87]	2024	75.00
Morigi et al. [88]	2020	87.50
Khurana et al. [89]	2023	100.00
Mukhi et al. [90]	2023	87.50
Li et al. [91]	2025	87.50
Lv et al. [92]	2024	62.50
Shi et al. [93]	2020	56.25
Salvatore et al. [94]	2014	81.25
Liu et al. [95]	2024	56.25
Su et al. [96]	2010	81.25
Motrapu et al. [97]	2020	68.75
Rosenbloom et al. [98]	2024	37.50
Minakawa et al. [99]	2019	68.75
Sawada et al. [100]	2023	81.25
Sharma et al. [101]	2016	56.25
Sunilkumar et al. [102]	2025	62.50
Lu et al. [103]	2024	62.50
Petrica et al. [104]	2023	100.00
Boi et al. [105]	2025	56.25
Song et al. [13]	2019	62.50
Qin et al. [106]	2020	56.25
Vestra et al. [107]	2003	81.25
Tian et al. [108]	2020	37.50
Veron et al. [109]	2021	62.50
Su et al. [110]	2022	62.50
Suarez et al. [111]	2024	56.25
Song et al. [112]	2022	62.50
Sun et al. [113]	2023	56.25
Stefansson et al. [114]	2022	87.50
Woo et al. [115]	2020	62.50
Sun et al. [116]	2025	62.50
Tao et al. [117]	2022	56.25
Uil et al. [118]	2021	87.50
Yang et al. [119]	2023	62.50
Ward et al. [120]	2025	31.25
Yamaguchi et al. [121]	2009	81.25
Yao et al. [122]	2020	56.25
Li et al. [123]	2020	62.50
Zeng et al. [124]	2023	100.00
Xue et al. [125]	2020	62.50
Zeng et al. [126]	2023	93.75
Wang et al. [127]	2021	62.50
Zhang et al. [14]	2016	56.25
Wu et al. [128]	2025	56.25
Yu et al. [129]	2022	62.50
Sawada et al. [3]	2016	81.25
Li et al. [130]	2024	93.75
Wang et al. [131]	2019	62.50
Pan et al. [9]	2018	56.25
Xu et al. [132]	2025	56.25
Zhang et al. [133]	2021	56.25
Wang et al. [134]	2024	62.50
Sawai et al. [135]	2006	81.25
Zhou et al. [136]	2019	62.50
Zhao et al. [137]	2023	87.50
Zhang et al. [138]	2024	62.50
Zhang et al. [139]	2024	62.50
Zhu et al. [140]	2025	62.50
Zhang et al. [141]	2025	62.50
Zhou et al. [142]	2024	87.50
Zhu et al. [143]	2021	62.50
Zuo et al. [144]	2024	62.50
Mean		81.25
SD		18.84
CV—%		23.19

% = percentage. SD = standard deviation. CV = coefficient of variation.

**Figure 1 ijms-26-08990-f001:**
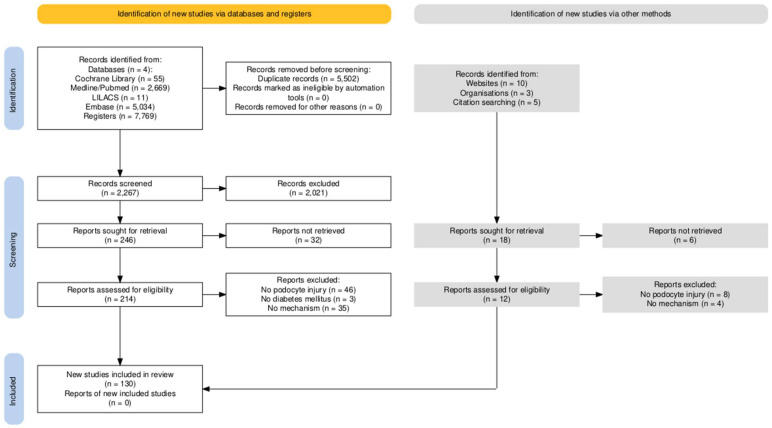
Flowchart illustrating the selection process of eligible studies. Generated using the PRISMA2020 tool [145].

Here, we summarize the annual output of studies on podocyte injury in diabetes mellitus.

Between 2002 and 2018, yearly output was intermittent and low (≤4 studies/year), totaling 23 studies across 13 reporting years. From 2019 onward, the field expanded markedly: 6 studies in 2019 (4.62%), followed by a peak of 24 in 2020 (18.46%). Subsequent years remained high: 18 in 2021 (13.85%), 9 in 2022 (6.92%), 16 in 2023 (12.31%), 19 in 2024 (14.62%), and 15 in 2025 (11.54%). Overall, 2019–2025 accounts for 107 of 130 studies (82.3%). Year-over-year changes mirror this surge: +50.00% in 2019 (vs. 2018), +300.00% in 2020 (vs. 2019), followed by contractions in 2021 (−25.00%) and 2022 (−50.00%), then renewed growth in 2023 (+77.78%) and 2024 (+18.75%), with a decline in 2025 (−21.05%). These patterns are depicted in Figure 2.

Trend analyses corroborate a strong temporal increase. The Spearman rank correlation between year and number of studies was ρ = 0.8652 (95% CI, 0.6773–0.9471; *p* < 0.0001), indicating a robust monotonic rise. Consistently, simple linear regression estimated an average increment of 0.81 studies per year (slope = 0.8115; 95% CI, 0.4581–1.165), with the model explaining 56.39% of the variance (R^2^ = 0.5639; F(1,18) = 23.28; *p* = 0.0001; residual SD = 5.084). The best-fit line was Y = 0.8115 × Year − 1628, reinforcing that publication volume has increased substantially over time. Collectively, these results indicate a pronounced and statistically significant acceleration of the field in the past decade.

Across all 130 eligible studies, human investigations were the single largest category (37/130; 28.5%), followed closely by in vitro + animal (28/130; 21.5%) and animal-only designs (27/130; 20.8%) (Figure 3A). Mixed-model designs were frequent: human + animal (10/130; 7.7%), human + in vitro + animal (10/130; 7.7%), and human + in vitro (8/130; 6.2%). In vitro-only studies were less common (7/130; 5.4%), while in silico appeared rarely (2/130; 1.5%); one report was unclassified (0.8%). Aggregating by component, any human element featured in 65/130 (50.0%), any animal element in 75/130 (57.7%), and any in vitro element in 53/130 (40.8%).

The temporal composition of study models (Figure 3B) shows three phases. Early phase (2002–2013): output was sparse and dominated by human-only designs (e.g., 2002, 2003, 2006, 2008, 2009, 2012, 2013 each 100% human); one exception was animal-only in 2011 (100%).

Diversification phase (2014–2018): mixed models emerge (e.g., in vitro + animal constitutes 100% in 2014; 2015–2018 introduce human + in vitro, human + animal, and human + in vitro + animal in varying proportions).

Expansion and consolidation (2019–2025): with the surge in publications (see Figure 2), portfolios broaden. In 2019, mixed designs predominate (72.7% of that year’s studies), led by human + animal (27.3%) and in vitro + animal/human + in vitro + animal (each 18.2%). In 2020, animal-only peaks (42.9%), with in vitro + animal close behind (33.3%), reflecting a shift toward mechanistic experimentation during the publication peak. The balance swings back toward human participation in 2021 (human-only 40.0%; human + in vitro + animal 26.7%) and remains mixed thereafter: 2022 distributes across in vitro + animal (33.3%), animal (25.0%), human (25.0%), in vitro (16.7%); 2023–2024 see renewed human-only leadership (35.3% and 42.1%, respectively) with persistent in vitro + animal contributions (23.5% and 26.3%). In silico analyses first appear in 2024 (5.3%) and rise modestly in 2025 (7.7%). In 2025 (data through July), animal-only (30.8%) and human + animal (23.1%) together comprise just over half of the yearly output, alongside in vitro + animal (15.4%), human-only (15.4%), and human + in vitro + animal (7.7%).

Together, these patterns indicate a clear evolution from early, single-modality human descriptions toward integrated, multi-model approaches that combine clinical relevance with mechanistic depth, a balance that has stabilized since 2019 while retaining year-to-year variability (Figure 3A,B).

The heatmap (Figure 4A) shows a clear continental gradient: Asia accounts for 60.0% of the corpus (78/130), followed by Europe 21.5% (28/130), North America 16.9% (22/130), and South America 1.5% (2/130). Within Asia, the most frequent designs were in vitro + animal (20) and human-only (19), with substantial use of animal-only (10) and mixed modalities, human + animal (8), human + in vitro + animal (9), and human + in vitro (6). In silico studies appeared exclusively in Asia (2), and the only unclassified record also originated there (1). Europe and North America emphasized human (10 and 7, respectively) and animal (9 and 8) designs; in vitro + animal was present in both (Europe 4; North America 4), whereas mixed three-arm designs were uncommon (Europe 1; North America 0). South America contributed two studies (human; human + animal).

Beyond the continental view presented in Figure 4, the country-level pattern is highly concentrated. China contributes the plurality of records across 2009–2025, with visible clusters in 2019–2020 and 2024–2025; Japan is another recurrent Asian contributor (2006, 2008–2009, 2016, 2019–2022, 2023). Outside Asia, the United States appears frequently (2013, 2014, 2020–2024, 2025), often leading North American output. In Europe, Germany and Italy recur (2011–2016, 2018, 2020–2025), alongside Romania, Poland, Denmark, Finland, Switzerland, France, Croatia, the United Kingdom, and The Netherlands (Amsterdam). Turkey also appears in 2025. In South America, Brazil contributes (2019, 2023), and there is at least one cross-continental collaboration (Chile–Spain, 2023). Additional multinational efforts include China–USA, Italy–USA, Switzerland–USA, and Europe–USA, underscoring the field’s collaborative character.

Alluvial flows (Figure 4B) reveal how these patterns unfolded over time. Notable concentrations include Asia: in vitro + animal in 2020 and 2024 (each n = 5) and Asia: human in 2024 (n = 5), alongside a 2021 cluster of Asia: human + in vitro + animal (n = 4). Mechanistic surges in 2020 also involved Europe: animal (n = 4) and North America: animal (n = 3). Human-focused activity intensified in Europe 2023 (n = 3) and North America 2024 (n = 3). In silico first appears in Asia 2024–2025 (each n = 1). For 2025 (indexed through July), Asia maintains emphasis on animal (n = 3) and human + animal (n = 3). Overall, these results indicate an Asian pivot toward integrated, multi-model experimentation since 2019, with Europe/North America sustaining strong human/animal emphases and South America remaining underrepresented.

A convergent picture emerges from the eligible studies (mapped in Table 2), in which chronic hyperglycemia, hemodynamic overload, lipid toxicity, sterile inflammation, and epigenetic/post-transcriptional reprogramming drive podocyte dedifferentiation, detachment, and death, culminating in proteinuria and progressive renal decline. Mechanistic nodes and their morphologic or molecular readouts are tightly coupled, and where available, quantitative signals (e.g., inverse correlations between podocyte density and albumin excretion) reinforce causality.

Filtration barrier and adhesion failure. The slit diaphragm is an early casualty: nephrin downregulation increases glomerular permeability and tracks with albuminuria in humans; ACE inhibition partially restores nephrin and reduces albuminuria, linking mechanism to clinical phenotype [60]. Cytoskeletal and junctional scaffolds are concurrently compromised: α-actinin-4 loss correlates with podocyte dysfunction/proteinuria [49], while loss of uniform connexin-43 under hyperglycemia associates with reduced renal function, implying disrupted intercellular coupling at the slit membrane [135]. Structural and quantitative abnormalities appear early and scale inversely with albumin excretion, podocyte density falls as AER rises, implicating podocyte loss as a primary driver of albuminuria [107]; human biopsy series corroborate fewer podocytes with more proteinuria and compensatory hypertrophy [75,96]. An FSP1/Snail1/ILK/TGF-β1 program indicates EMT-like reprogramming in podocytes, mechanistically tied to GBM detachment and worse clinico-pathologic DN [121]. Adhesion dynamics are phase-dependent: α3β1-integrin is induced in early DN, facilitating foot-process effacement/detachment via TGF-β1 signaling [3], whereas late disease features reduced integrin tone with broadened processes [142]. Two amplifiers further erode the diaphragm: hyperglycemia accelerates dynein-dependent nephrin degradation (DynII1/DCTN1) [146], and anti-nephrin autoantibodies add an immune hit with ATP-depletion-related cytoskeletal injury [68]. Cell-intrinsic guardians and liabilities add texture: BASP1 co-represses WT1 to activate p53-mediated apoptosis [133], CKAP4 maintains actin–microtubule architecture (its loss precipitates effacement/detachment) [105], and SRGAP2a restrains RhoA/Cdc42 to curb motility and preserve structure under hyperglycemic/TGF-β stress [87]. A compensatory rise in podocyte ClC-5 in proteinuric patients suggests augmented albumin endocytosis within the injured barrier [26].

mTOR–stress signaling and epitranscriptomic control. Podocyte mTORC1 is a bidirectional hazard: hyperactivation dislocates slit-diaphragm proteins, induces EMT-like and ER-stress signatures, and leads to podocyte loss, mesangial expansion, and proteinuria [58]; conversely, deleting or broadly dysregulating mTORC1 yields hypertrophy, effacement, detachment, and suppressed autophagy, underscoring the need for tight rheostasis [59]. Fine-tuning layers align mechanism with lesion: miR-99a-5p constrains mTOR/EMT [118]; REDD1 lowers nephrin/podocin and heightens TRPC6-Ca^2+^ influx, disorganizing the cytoskeleton [102]; Rheb1 loss accelerates mitochondrial senescence via Atp5f1c acetylation independently of mTORC1 [103]. Epitranscriptomic “writers” shape vulnerability: METTL14-mediated m6A reduces Sirt1 and promotes podocyte injury [70], while METTL3 stabilizes TIMP2 (via IGF2BP2) to activate Notch-inflammation/apoptosis and, in separate work, engages an MDM2/Notch axis in dedifferentiation [57,128]. Upstream, TXNIP links hyperglycemia to oxidative stress and mTOR/EMT activation; knockdown reduces ROS and renal injury and associates with mTOR activity in human biopsies [13]. EGFR signaling elevates Rubicon and blocks autophagy through mTOR–p70S6K/RPS6, worsening podocyte injury [123]. ERK activation demonstrated in human DN podocytes further implicates VEGF/ribosome-biogenesis programs [21]. Clinically relevant counter-signaling is possible: saxagliptin suppresses p38 activity while increasing nephrin/podocin independently of glycemia [147].

Lysosomal dysfunction impairs albumin degradation, escalates cytokines, and drives glomerulosclerosis, directly connecting protein overload to scarring [23]. High-glucose mesangial signals suppress podocyte ERAD and nephrin phosphorylation, anatomically embedding the lesion in intra-glomerular crosstalk [36]. Therapeutically, HGF restores autophagy/lysosomal flux via PI3K/Akt–GSK3β–TFEB, reducing albuminuria and podocyte loss [41]. UCP2 supports macroautophagy (loss worsens proteinuria/injury) [119]. DOT1L–PLCL1 improves lipid handling and reduces lipotoxic damage [53], and BTG2 couples mTORC1 inhibition to pro-autophagic, anti-inflammatory effects [87].

Palmitate elicits an ultimately “futile” antioxidant response and oxidative ultrastructural damage [78]. Ceramides accumulate and injure mitochondria; CerS6–VDAC1 interaction triggers mtDNA leakage and cGAS–STING activation [115,140]. Insufficient FAO and mitochondrial dysfunction sustain vulnerability [148], exacerbated by PGRN deficiency with failure of the PGRN–Sirt1–PGC-1α/FoxO1 program [136]. Protective metabolic levers include β-hydroxybutyrate (GSK3β inhibition, Nrf2 activation, less senescence) [149]; SGLT2i-driven ERRα–ACOX1 activation (more FAO, less lipotoxicity, structural repair) [72]; and GM3 restoration by valproate [86]. Lipid-handling regulators add causality: CCDC92 promotes lipid deposition via ABCA1 [144], and ACSS2 epigenetically activates mTORC1 and suppresses autophagy [80]. At the calcium–mitochondria interface, TRPC6-dependent Ca^2+^ influx activates calpain-1/CDK5/Drp1 to drive mitochondrial fission and apoptosis [129], while Orai1-mediated SOCE activates calpain and disorganizes F-actin/nephrin [117]. Complement C3a–C3aR further disrupts podocyte bioenergetics; antagonism restores mitochondrial function/density [88]. Systemic mtDNA/OXPHOS injury and ROS parallel glomerulotubular inflammation even in normoalbuminuric DKD [104].

Podocytes upregulate CD80/B7-1 under high glucose (PI3Kα), disrupting the cytoskeleton and inducing apoptosis; CTLA4-Ig reverses this phenotype [39]. High glucose/AGEs/ROS activate NLRP3, engaging canonical and non-canonical arms that propagate podocyte dysfunction [55]; blocking caspase-1 with carnosine attenuates pyroptotic injury [150]. TRAIL–DR5 signaling triggers PANoptosis, and TRAIL/DR5 deletion reduces glomerular damage [92]. Complement disinhibition is pathogenic: DAF/CD55 loss unleashes C3 convertase, activating C3a/C3aR and an IL-1β/IL-1R1 loop that lowers nephrin and remodels actin; low DAF with C3d positivity and high urinary C3a coincide with FSGS-like lesions and proteinuria [33]. Additional nodes, HDAC4 → calcineurin apoptosis [93], ROCK-mediated mesangial fibrosis and podocyte apoptosis [84], RARRES1 → p53 apoptosis [32], RIPK3-driven NF-κB inflammation independent of necroptosis [69], and DHAP-mTORC1/ROS/NLRP3 coupling [151], further knit mechanism to lesion. In PGNMID, endothelial PV-1 overexpression links complement/IgG deposition to oxidative, inflammatory crosstalk that secondarily injures podocytes [100]. GH-TGF-β1/Notch signaling drives podocyte binucleation/mitotic catastrophe; blocking GHR/TGFBR1 prevents these cytologic lesions and DN features [82,90].

Obliterative microangiopathy (arteriolosclerosis, hyalinosis) and glomerular ischemia appear in DN and associate with collapsing glomerulopathy, tuft collapse, epithelial proliferation, VEGF overexpression, and poor outcomes/ESRD [94]. Early hyperfiltration couples to podocyte depletion and GBM thickening; endothelial stress with mesangial crosstalk activates fibrosis programs that mirror advancing histologic class [50,114]. Hyperglycemia activates the polyol/cPKC axes and podocyte loss, while DGKα/67LR maintains adhesion [31]. PECs enlarge Bowman’s capsule ECM in human DN [43] and, under severe microvascular hypoxia, associate with extracapillary hypercellularity and loss of podocyte phenotype [42]. Diabetic co-culture models confirm that HG/MGO deform GEC–podocyte transcriptomes and degrade ECM/barrier properties [34]. Uremic gut-derived metabolites and loss of retinoic-acid signaling tie systemic milieu to glomerular/endothelial dysfunction [19].

Canonical DN histology, mesangial expansion, Kimmelstiel–Wilson nodules, podocyte depletion, remains central [50]. Spatial metabolomics (MALDI-IMS/MxIF) maps lipid signatures that co-localize with podocyte loss and mesangial expansion, providing tissue-level surrogates of injury [28]. Urinary nephrin detects early podocyte injury [52]; elevated urinary podocin and intrarenal podocalyxin predict progression and track function [126]; the urinary podocin:nephrin mRNA ratio correlates with tubulointerstitial fibrosis [130]. Clinicopathologic resources ground these signals: in TRIDENT, eGFR correlates most with interstitial fibrosis and glomerular epithelial changes [85]; the nPOD-K biobank anchors histologic trajectory studies [120]. Therapeutically, pathway-directed interventions show structural dividends: HGF and DOT1L/PLCL1/BTG2 restore autophagy/lipid handling [41,53]; CTLA4-Ig, C3aR antagonism, DR5/TRAIL blockade, and inflammasome targeting blunt immune lesions [39,55,88,92]; metabolic correction with SGLT2i (reduced MAMs, AMPK activation), β-hydroxybutyrate, and VPA/GM3 improves ultrastructure [72,81,86,149]; and systemic milieu shifts after Roux-en-Y gastric bypass reverse podocyte dedifferentiation/effacement alongside reduced albuminuria [27].

Taken together, the field supports a staged lesion cascade, slit-diaphragm/adhesion failure (nephrin/integrins/connexins/trafficking), mTOR–autophagy–ER/mitochondrial disequilibrium, lipotoxicity with defective FAO and oxidative stress, immune/complement/inflammasome amplification, and epigenetic–transcriptomic reprogramming via m6A, ncRNAs, and APA, modulated by hemodynamics and endothelial/mesangial crosstalk. Each node is paired with a reproducible description at the tissue or molecular level and, in several instances, with therapeutic reversibility. For a study-by-study mapping of mechanism to lesion descriptor, see Table 2 (mechanisms and descriptions), which underpins the narrative links and citations summarized here.

**Table 2 ijms-26-08990-t002:** Summary of inducing mechanisms and main characteristics of lesions associated with podocyte damage related to diabetes mellitus.

Autor (Year)	Mechanism of Lesion	Description
Langham et al., 2002 [60]	Reduced nephrin expression and glomerular permeability.	Nephrin downregulation increases glomerular permeability and is associated with proteinuria. ACE inhibition restores nephrin levels and mitigates albuminuria, indicating nephrin’s protective role in diabetic nephropathy.
Vestra et al., 2003 [107]	Mesangial expansion and podocyte injury	Podocyte density and structure are altered in type 2 diabetes, contributing to albuminuria. Podocyte density is inversely related to AER, and structural changes occur early in DN progression, indicating podocyte loss as a key pathogenic factor.
Sawai et al., 2006 [135]	Downregulation and heterogeneity of Cx43 in podocytes under hyperglycemia impairs intercellular communication and slit diaphragm integrity.	Loss of uniform Cx43 expression correlates with decreased renal function, implicating its role in diabetic nephropathy progression.
Kimura et al., 2008 [49]	Low alpha-actinin-4 expression damages podocytes and slit diaphragms, contributing to proteinuria.	Alpha-actinin-4 downregulation correlates with podocyte dysfunction and proteinuria severity in human diabetic kidneys.
Miyauchi et al., 2009 [75]	Podocyte loss is inversely correlated with proteinuria; hypertrophy compensates for podocyte loss due to glomerular pressure.	Human renal biopsy study shows reduced podocyte number correlates with increased proteinuria and hypertrophy. ACE-Is/ARBs included as treatment variables but their mechanistic role in podocyte hypertrophy is unclear.
Su et al., 2010 [96]	Reduced podocyte number and density in DN correlates inversely with proteinuria severity, involving WT1 changes and cytoplasmic structural alterations.	DN patients show decreased podocyte density and cytoplasmic coverage, with changes correlating with proteinuria levels. WT1 is used as a lesion marker.
Yamaguchi et al., 2009 [121]	Podocyte detachment from the glomerular basement membrane via EMT induced by FSP1, Snail1, ILK, and TGF-β1.	FSP1 expression in podocytes is associated with EMT markers and more severe clinical and pathological manifestations in diabetic nephropathy.
Inoki et al., 2011 [58]	mTORC1 hyperactivation.	Excessive mTORC1 activity in podocytes disrupts slit diaphragm protein localization, promotes epithelial–mesenchymal transition-like changes, induces ER stress, and leads to podocyte loss, mesangial expansion, and proteinuria in DN.
Gödel et al., 2011 [59]	mTOR dysregulation and podocyte stress.	Both overactivation and deletion of mTORC1 in podocytes lead to injury. mTOR signaling affects hypertrophy, foot process effacement, detachment, and autophagy suppression. Balanced mTOR activity is essential for podocyte homeostasis.
Ceol et al., 2012 [26]	ClC-5 overexpression in podocytes enhances albumin endocytosis, potentially compensating protein overload in nephropathies.	The study demonstrates ClC-5 is overexpressed in podocytes of proteinuric patients, supporting its role in albumin endocytosis.
Salvatore et al., 2014 [94]	Ischemic podocyte injury due to obliterative microvascular disease (arteriolosclerosis, hyalinosis), contributing to collapsing glomerulopathy (CG) in DN.	CG in DN is characterized by glomerular tuft collapse and epithelial proliferation, with loss of podocyte markers and VEGF overexpression. It is linked to poor prognosis and progression to ESRD.
Carson et al., 2014 [23]	Impaired lysosomal degradation in podocytes leads to albumin accumulation, cytokine production, and glomerulosclerosis.	Lysosomal dysfunction in podocytes impairs albumin processing, increases cytokine production and promotes glomerulosclerosis.
Fiorina et al., 2014 [39]	High glucose induces podocyte CD80/B7-1 via PI3Kα, leading to cytoskeleton disruption and apoptosis; reversed by CTLA4-Ig.	CD80/B7-1 upregulation mediates diabetic podocyte damage and albuminuria; blockade via CTLA4-Ig shows therapeutic potential.
Ivanac-Janković et al., 2015 [22]	Downregulation of BMP-7 in DN enhances TGF-β-driven fibrosis and reduces podocyte survival.	The study shows BMP-7 expression decreases in advanced DN stages, highlighting its protective role against inflammation and fibrosis.
Holderied et al., 2015 [43]	Diabetic-induced activation of PECs increases ECM secretion, thickening Bowman’s capsule under hyperglycemic and AGE conditions.	Diabetic PECs promote ECM expansion of Bowman’s capsule, worsening glomerular sclerosis in the absence of TGF-b1 autocrine feedback.
Sharma’s et al., 2016 [101]	MDM2 downregulation disrupts podocyte and tubular function, impairing metabolic pathways and reducing recovery capacity from injury.	MDM2 is reduced in DN and associated with metabolic dysfunction. Its loss leads to altered protein interactions and reduced renal resilience.
Zhang et al., 2016 [14]	Wnt/β-catenin signaling upregulates UCH-L1, altering podocyte morphology and increasing motility, contributing to DN.	High glucose increases UCH-L1 via Wnt/β-catenin signaling in podocytes; UCH-L1 is a potential therapeutic target in DN.
Sawada et al., 2016 [3]	Upregulation of α3β1-Integrin in podocytes contributes to foot process effacement and detachment via TGF-β1 signaling.	α3β1-Integrin is upregulated in early DN stages, facilitating detachment and cytoskeletal changes in podocytes.
Zhou et al., 2017 [20]	High glucose-induced miR-27a upregulation suppresses PPARγ, activates β-catenin, promotes mesenchymal transition and podocyte apoptosis.	miR-27a expression is stimulated by hyperglycemia, suppressing PPARγ and activating β-catenin, leading to podocyte injury and renal dysfunction in diabetic rats.
Bai et al., 2018 [61]	LINC01619/miR-27a/FOXO1 axis dysregulation.	Downregulation of LINC01619 removes its sponge effect on miR-27a, leading to suppression of FOXO1 and induction of ER stress. This enhances oxidative stress, apoptosis, and foot process effacement in podocytes.
Endlich et al., 2018 [35]	BDNF knockdown reduces nephrin/podocin, alters glomerular morphology, induces podocyte dedifferentiation and developmental defects.	Zebrafish and human data confirm BDNF’s role in podocyte function and as a biomarker for glomerular injury.
Lee et al., 2018 [78]	Palmitic acid induces mitochondrial ROS and reduces antioxidant enzymes, exacerbating oxidative stress in podocytes.	PA increases antioxidant proteins transiently but causes long-term suppression, correlating with oxidative damage in advanced DN.
Canney et al., 2020 [27]	Podocyte dedifferentiation and foot process effacement reversed by Roux-en-Y gastric bypass via improved glucose control.	Post-RYGB, patients showed reduced albuminuria and improved podocyte structure, associated with enhanced metabolic control.
Hu et al., 2020 [37]	Saxagliptin inhibits renal p38MAPK and enhances nephrin/podocin expression, independently of glucose-lowering effects.	Saxagliptin reduces renal injury by modulating nephrin/podocin expression and suppressing inflammation-related signaling.
Hu et al., 2023 [48]	LRH-1/GLS2 downregulation in podocytes impairs glutaminolysis, causing mitochondrial dysfunction and apoptosis.	LRH-1 loss impairs glutamine metabolism, driving podocyte apoptosis; restoring LRH-1 mitigates DKD injury.
Liang et al., 2020 [71]	GSK3β overactivity and ECM accumulation.	GSK3β is overexpressed in DKD, promoting ECM accumulation and podocyte injury. Its urinary activity predicts disease progression, serving as a potential biomarker.
Minakawa et al., 2019 [99]	Glomerular enlargement in early T2DM causes podocyte hypertrophic stress, detachment, and loss, leading to albuminuria.	In Zucker rats, glomerular volume increase precedes albuminuria, correlating with podocyte depletion. Detachment is driven by IGF1/2 and mTORC1 activation.
Song et al., 2019 [13]	TXNIP induction via high glucose.	Thioredoxin-interacting protein (TXNIP) is induced by high glucose and promotes oxidative stress by inhibiting thioredoxin. TXNIP knockdown disrupts EMT, reduces ROS, and inhibits mTOR pathway. In diabetic mice, TXNIP deficiency alleviates renal damage, and its expression correlates with mTOR activation in DN biopsies.
Qin et al., 2019 [106]	Mitochondrial dysfunction and impaired FAO.	Berberine improves insulin sensitivity and glucose tolerance, reduces albuminuria, and activates AMPK and PGC-1α pathways. These regulate mitochondrial energy homeostasis, fatty acid oxidation, and protect podocytes from oxidative stress in diabetic kidney disease.
Woo et al., 2020 [115]	Ceramide accumulation and mitochondrial injury.	Ceramide accumulation induces ROS-mediated mitochondrial damage in podocytes. Myriocin reduces ceramide synthesis and protects against DN progression.
Uil et al., 2021 [118]	miR-99a-5p regulation of mTOR and EMT.	miR-99a-5p inhibits mTOR and vimentin, protecting podocytes from EMT and injury in DN. Identified in extracellular vesicles from patients with micro/macroalbuminuria.
Wang et al., 2019 [131]	Downregulation of miR-27a/b increases FOXO1 expression, enhancing PEPCK and G6pase, leading to hepatic gluconeogenesis and hyperglycemia.	miR-27a/b regulates hepatic gluconeogenesis by targeting FOXO1. Overexpression reduces glucose output, suggesting a therapeutic role in type 2 diabetes.
Zhou et al., 2019 [136]	PGRN deficiency leads to mitochondrial dysfunction in podocytes by disrupting PGRN-Sirt1-PGC-1α/FoxO1 signaling, impairing mitophagy and biogenesis.	PGRN maintains mitochondrial homeostasis in podocytes. Its deficiency exacerbates injury, while rPGRN treatment restores function.
Shetty et al., 2021 [25]	Podocyte and tubular injury due to viral infection or systemic inflammation, with APOL1 genotypes increasing susceptibility.	Podocytopathy and protein overload tubulopathy were observed in COVID-19 patients, with APOL1 genotypes potentially influencing severity.
Denhez et al., 2020 [29]	Palmitate-induced FFA triggers IKKβ and mTORC1 activation, promoting insulin resistance in podocytes.	Palmitic acid induces podocyte insulin resistance via ceramide production and serine 307 phosphorylation of IRS1.
Audzeyenka et al., 2020 [30]	Cathepsin C overexpression in podocytes causes cytoskeletal disruption and insulin resistance under hyperglycemia.	Cathepsin C expression is increased in diabetic conditions, altering podocyte cytoskeleton and promoting albumin leakage.
Hayashi et al., 2020 [31]	Glomerular damage due to hyperglycemia-induced microvascular disorders, polyol pathway activation, cPKC activation, and podocyte loss; DGKα/67LR interactions maintain adhesion.	EGCg activates DGKα to maintain glomerular integrity and podocyte adhesion, mitigating diabetic nephropathy progression.
Chen et al., 2020 [32]	RARRES1 overexpression triggers podocyte apoptosis via RIOK1 interaction and p53 activation.	RARRES1 is upregulated in DN and induces podocyte injury via p53 signaling, supported by murine and human biopsy findings.
Fujimoto et al., 2020 [36]	High glucose suppresses podocyte ERAD pathway via mesangial crosstalk, leading to ER stress and nephrin phosphorylation suppression.	High-glucose-induced mesangial signals disrupt ERAD in podocytes, contributing to progressive DN and proteinuria.
Hou et al., 2020 [41]	HGF modulates PI3K/Akt-GSK3β-TFEB signaling to restore podocyte autophagy and lysosomal function.	HGF restores autophagy in diabetic podocytes, preserving function through PI3K/Akt-GSK3β-TFEB axis modulation.
Jiang et al., 2020 [46]	Smad3 activation under hyperglycemia disrupts cytoskeleton via transgelin and caspase-3, leading to podocyte damage.	Smad3 signaling under diabetic stress promotes actin remodeling and transgelin expression, compromising podocyte structure.
Fu et al., 2020 [47]	JAML activation impairs lipid metabolism in podocytes via SIRT1-SREBP1 and AMPK signaling.	JAML exacerbates podocyte lipid imbalance and injury through dysregulated SIRT1-mediated transcription and SREBP1 acetylation.
Wang et al., 2020 [62]	miR-770-5p-mediated TIMP3 suppression.	miR-770-5p upregulation promotes podocyte apoptosis and inflammation by targeting TIMP3, a protective factor. Its depletion reduces these effects, indicating a role in DN pathogenesis and a potential therapeutic target.
Lai et al., 2020 [63]	BAMBI deletion and TGF-β pathway overactivation.	Loss of BAMBI enhances TGF-β signaling. In podocytes, ALK5/Smad2/3 activation leads to apoptosis and loss. In endothelial cells, ALK1/Smad1/5 promotes proliferation and vascular dysfunction, contributing to DN progression.
Kostic et al., 2020 [65]	Tubulointerstitial fibrosis and mitochondrial damage.	CKD progression involves oxidative stress, glomerulosclerosis, and AIF-related pathways mediating apoptosis and cell survival. AIF is upregulated in glomeruli of diabetic rats, suggesting its role as a biomarker.
Liebisch et al., 2020 [66]	Epigenetic alterations due to AGEs.	AGEs reduce NIPP1 and EZH2 expression, decreasing H3K27me3 and promoting transcription of pro-disease genes in podocytes, contributing to DKD and metabolic memory.
Morigi et al., 2020 [88]	Complement activation via C3a causes podocyte mitochondrial damage and dysfunction, leading to proteinuria.	C3aR blockade restores mitochondrial function and podocyte density, offering therapeutic potential in DN.
Shi et al., 2020 [152]	Hyperglycemia upregulates HDAC4, which enhances calcineurin (CaN) signaling and promotes podocyte apoptosis. HDAC4 knockdown reduces CaN expression and apoptosis.	HDAC4 mediates CaN signaling in high-glucose conditions, leading to podocyte apoptosis. Its silencing attenuates this effect, suggesting a key role in DN pathophysiology.
Motrapu et al., 2020 [97]	Podocyte loss due to increased shear stress from RAS and SGLT2 pathways; MRE therapy increases podocyte density, further improved by BIO enhancing filtration slit density.	BIO combined with MRE attenuates CKD progression in diabetic mice by restoring podocyte numbers and filtration structure, improving GFR.
Tian et al., 2020 [108]	Loss of GAK and calpain activation.	Podocyte-specific GAK deficiency leads to albuminuria and glomerulosclerosis. Increased intracellular calcium activates calpain-1/2, causing NF-κB activation and GADD45B expression. Calpain inhibition mitigates these effects.
Yao et al., 2020 [122]	circ_0000285 sponges miR-654-3p, activating MAPK6 and promoting podocyte injury.	circ_0000285 promotes diabetic nephropathy progression via miR-654-3p suppression and MAPK6 activation.
Li et al., 2021 [123]	Activation of EGFR signaling increases rubicon expression and inhibits autophagy through mTOR-p70 S6K and RPS6 pathways, contributing to podocyte injury.	EGFR deletion in podocytes enhances autophagy and reduces albuminuria and inflammation, highlighting its role in DN progression.
Xue et al., 2020 [125]	Modulation of the PTEN-PDK1-Akt-mTOR pathway and reduction in Nox4-driven ROS production improves podocyte viability.	Xuesaitong treatment in diabetic rats reduces albuminuria and podocyte apoptosis via PTEN pathway regulation and oxidative stress mitigation.
Wang et al., 2021 [127]	Cdk5 upregulation under diabetic conditions leads to synaptopodin/nephrin downregulation, ROS increase, mitochondrial fission, and ATP depletion.	Cdk5-mediated mitochondrial dysfunction contributes to podocyte injury; inhibition of Cdk5 improves mitochondrial function and protects podocytes.
Cao et al., 2021 [45]	Diminished DACH1 reduces PTIP recruitment, increases H3K4Me3, and enhances gene transcription linked to podocyte vulnerability.	Loss of DACH1 epigenetically activates injury pathways; restoring DACH1 protects podocytes against DKD-related stress.
Kondapi et al., 2021 [52]	Hyperglycemia induces nephrin excretion, podocyte foot process retraction, cytoskeletal rearrangement, and glomerular/tubular thickening.	Urinary nephrin correlates with podocyte damage and is a sensitive early biomarker of diabetic nephropathy.
Kondapi et al., 2021 [54]	Podocyte structural damage and RAAS activation.	Podocyte injury in DKD involves foot process effacement, hypertrophy, apoptosis, and detachment. Disruption of slit diaphragm proteins (e.g., podocin) and RAAS activation with angiotensin II signaling further promote proteinuria and glomerulosclerosis.
Kawaguchi et al., 2021 [56]	Parietal epithelial cell (PEC) hypertrophy and injury.	High-glucose exposure leads to PEC hypertrophy, vacuolization, S-phase arrest, and mitotic catastrophe. These changes impair glomerular regeneration and may induce periglomerular inflammation, contributing to glomerular injury in DN.
Liu et al., 2022 [67]	Bcl-2-mediated regulation of apoptosis and autophagy.	Wogonin restores autophagy and inhibits apoptosis in podocytes via Bcl-2 interaction, reducing albuminuria and histological damage in diabetic mice.
Lu et al., 2021 [70]	METTL14-induced degradation of Sirt1.	METTL14 promotes m6A modification of Sirt1 mRNA, leading to podocyte injury. METTL14 deletion improves glomerular structure and reduces proteinuria.
Lu et al., 2021 [74]	GPR43-mediated insulin resistance.	GPR43 activation suppresses AMPKα via the PKC-PLC pathway, impairing podocyte insulin signaling. Its inhibition improves insulin sensitivity and reduces albuminuria.
Löwen et al., 2021 [79]	GBM component accumulation, endothelial proliferation, and podocyte death lead to glomerulosclerosis and interstitial fibrosis.	Glomerular vessel leakiness, tuft adhesions, and insudative damage propagate tubular degeneration and fibrosis in DN.
Nishad et al., 2021 [82]	Growth hormone induces podocyte cycle reentry via TGF-β1 and Notch, causing cytokinesis failure and cell death.	GH-driven TGF-β1/Notch signaling causes binucleation and mitotic catastrophe; inhibition prevents podocyte loss in DN.
Petrica et al., 2021 [83]	lncRNAs modulate podocyte injury by interacting with miRNAs and regulating oxidative stress and fibrogenesis.	lncRNAs like MALAT1 and NEAT1 worsen DKD by promoting inflammation, while MIAT and TUG1 offer protective effects.
Matoba et al., 2021 [84]	ROCK signaling mediates mesangial fibrosis, podocyte apoptosis, and inflammation, contributing to DKD.	Fasudil reduces proteinuria in diabetic patients by inhibiting ROCK without affecting blood pressure or eGFR.
Palmer et al., 2021 [85]	Glomerular epithelial hypertrophy and podocyte injury contribute to interstitial fibrosis and kidney function decline.	TRIDENT cohort shows eGFR correlates with interstitial fibrosis and glomerular epithelial changes in DKD patients.
Su et al., 2022 [110]	Risa overexpression and autophagy inhibition.	Risa, a long non-coding RNA, inhibits autophagy via Sirt1/GSK3Î^2^ axis, causing podocyte injury. Its suppression enhances autophagy and reduces injury, making it a potential therapeutic target.
Zhang et al., 2021 [133]	BASP1 acts as a cosuppressor of WT1, activating the p53 pathway and inducing podocyte apoptosis in diabetic nephropathy.	BASP1 is upregulated in diabetic nephropathy and promotes podocyte apoptosis via p53 activation, suggesting a role in disease progression.
Zhu et al., 2021 [150]	Caspase-1-mediated pyroptosis via GSDMD-N formation causes membrane rupture and cytokine release; carnosine inhibits this pathway.	Carnosine reduces inflammation and podocyte injury in DN by targeting caspase-1, suggesting its therapeutic potential.
Fang et al., 2021 [40]	β-hydroxybutyrate inhibits GSK3β, enhancing Nrf2 activity and reducing podocyte senescence and injury.	The ketone body β-hydroxybutyrate activates antioxidant pathways, reducing renal oxidative stress and podocyte aging.
Shahzad et al., 2022 [55]	NLRP3 inflammasome activation.	High glucose, AGEs, and ROS activate the NLRP3 inflammasome in podocytes, triggering canonical (caspase-1, IL-1β, IL-18) and non-canonical (autophagy regulation) pathways, contributing to sterile inflammation, podocyte dysfunction, and DKD progression.
Jiang et al., 2022 [57]	METTL3-mediated m6A modification and TIMP2 stabilization.	METTL3 enhances m6A modification of TIMP2 mRNA. IGF2BP2 binds to m6A sites, stabilizing TIMP2, which promotes Notch signaling, inflammation, and apoptosis in podocytes, leading to injury in DN.
Mukhi et al., 2023 [90]	GH induces TGF-β1 expression, activating SMAD signaling and increasing podocyte permeability.	GHR deletion or TGF-βR1 inhibition in podocytes prevents GH-induced SMAD activation and DN manifestations.
Sawada et al., 2023 [100]	PGNMID involves PV-1 overexpression in glomerular endothelial cells, triggering oxidative stress and inflammatory crosstalk to podocytes.	PV-1 expression correlates with podocyte injury in PGNMID. Complement activation and IgG deposition promote inflammation and podocyte damage.
Veron et al., 2021 [109]	VEGF-A knockdown and eNOS deficiency.	VEGF-A knockdown in eNOS-deficient mice induces diffuse glomerulosclerosis and proteinuria. S-nitrosylation of β3-integrin, laminin, and GSNOR contributes to renal damage. NO and thiol levels help protect renal function in diabetic mice.
Song et al., 2022 [112]	Sestrin2 and TSP-1/TGF-β1/Smad3 modulation.	Sestrin2 protects podocytes by modulating the TSP-1/TGF-β1/Smad3 pathway, reducing oxidative stress, phenotypic changes, and apoptosis in DKD.
Sun et al., 2023 [113]	Dynein-mediated nephrin degradation.	Hyperglycemia increases dynein expression, impairing nephrin trafficking and promoting degradation via DynII1 and DCTN1. This disrupts the kidney’s molecular sieve and contributes to DN.
Stefansson et al., 2022 [114]	Hyperfiltration and endothelial stress.	Hyperfiltration in early diabetes leads to podocyte depletion and GBM thickening. Endothelial stress response and mesangial cell crosstalk activate fibrosis-related pathways, contributing to DN.
Tao et al., 2022 [117]	Orai1-mediated SOCE and calpain activation.	Hyperglycemia induces SOCE via Orai1, activating calpain, which causes F-actin disorganization and nephrin loss, leading to podocyte injury.
Zeng et al., 2023 [126]	Podocyte damage and detachment from the GBM leads to urinary shedding of podocyte fragments and glomerulosclerosis.	Elevated urinary podocin and intrarenal podocalyxin levels predict DKD progression and correlate with kidney function decline.
Yu et al., 2022 [129]	TRPC6-mediated Ca^2+^ influx activates calpain-1, CDK5, and Drp1 phosphorylation, triggering mitochondrial fission in podocytes.	TRPC6 promotes podocyte mitochondrial dysfunction and apoptosis in diabetic conditions via the Ca^2+^/calpain-1/CDK5/Drp1 axis.
Balint et al., 2023 [19]	Endothelial dysfunction and BBB disruption due to gut-derived metabolites (e.g., indoxyl sulfate), oxidative stress, and loss of retinoic acid signaling.	This study used metabolomics of serum and urine in T2DM patients to identify early DKD biomarkers like indoxyl sulfate and all-trans retinoic acid, linked to podocyte and endothelial dysfunction.
Albrecht et al., 2023 [34]	HG and MGO disrupt GEC-podocyte crosstalk, impairing the filtration barrier and ECM structure; ID1/ID3 upregulation is insufficiently protective.	GEC-podocyte co-culture under diabetic stress reveals altered gene expression, confirming impaired intercellular signaling.
Chen et al., 2024 [38]	Renal inflammation and immune signaling cause podocyte injury; gene dysregulation (e.g., TGFBR3, PTGDS, FGF1/9) contributes to fibrosis and oxidative stress.	Bioinformatics revealed immune-related gene dysregulation and inflammatory mediators contributing to DKD pathology.
Hu et al., 2023 [51]	ENST00000436340 interacts with PTBP1 to degrade RAB3B mRNA, impairing cytoskeleton and GLUT4 translocation.	lncRNA ENST00000436340 disrupts cytoskeletal stability and glucose transport in podocytes, worsening DKD.
Martins et al., 2023 [76]	Hyperglycemia and angiotensin II lead to podocyte damage via actin destabilization, foot process effacement, and inflammation mediated by Mindin.	Mindin expression is elevated in DN and correlates with foot process effacement, suggesting its role as a biomarker of podocyte damage and chronic inflammation.
Lizotte et al., 2023 [77]	SHP-1 promotes podocyte injury by impairing slit diaphragm proteins and enhancing SUMO2 modification of podocin.	Podocyte-specific SHP-1 deletion in mice prevents albuminuria and structural damage in DKD, highlighting SHP-1 as a potential therapeutic target.
Lu et al., 2023 [80]	ACSS2 upregulation inhibits autophagy via mTORC1 activation, promoting podocyte injury and inflammation.	ACSS2 enhances raptor expression via histone acetylation, impairs autophagy, and contributes to DN progression.
Naito et al., 2023 [86]	Reduced GM3 in podocytes leads to albuminuria and glomerular lesions; VPA restores GM3 and reduces damage.	VPA-induced GM3 expression in podocytes mitigates podocyte loss and mesangial expansion in DN.
Khurana et al., 2023 [89]	Reduced DNA methylation at key sites affects gene expression related to insulin signaling and fibrosis.	Hypomethylation at CTCF/Pol2B sites in leukocytes from DN patients links to disease progression and renal decline.
Liu et al., 2024 [95]	Circ-0000953 modulates autophagy and inflammation by sponging Mir665-3p and regulating Atg4b in podocytes.	Circ-0000953 contributes to autophagy dysregulation in DN through the Mir665-3p-Atg4b axis, suggesting a regulatory role in podocyte injury.
Petrica et al., 2023 [104]	mtDNA damage and impaired OXPHOS lead to ROS overproduction, inflammation, and tissue injury at glomerular and tubular levels.	mtDNA alterations in blood/urine reflect inflammation in normoalbuminuric DKD. These changes are linked to podocyte and PT dysfunction.
Suarez et al., 2024 [111]	Impaired ENT2 activity and adenosine dysregulation.	In DN, insulin regulation of ENT2 is impaired, causing loss of adenosine homeostasis and glomerular alterations. Human podocyte and rat glomeruli models confirm ENT2 dysfunction in diabetes.
Yang et al., 2023 [119]	UCP2 deficiency impairs autophagy in podocytes by modulating mTORC1 phosphorylation and activating AMPK, which may inhibit the mTOR pathway.	UCP2 expression increases under diabetic conditions as a compensatory response. Its deficiency impairs autophagy, worsening podocyte injury and proteinuria, indicating its critical role in maintaining podocyte homeostasis.
Zeng et al., 2023 [124]	GSK3β overactivity induces dedifferentiation, ECM accumulation, and profibrotic cytokine expression, accelerating DKD.	Intrarenal and urinary GSK3β levels are elevated in DKD; the pY216-GSK3β/total GSK3β ratio correlates with disease progression.
Zhao et al., 2023 [137]	Alternative polyadenylation (APA) leads to 3′UTR lengthening, enhancing translation of inflammation-related proteins and activating ER stress and NF-κB signaling.	APA promotes diabetic nephropathy progression by increasing protein synthesis involved in inflammation and stress pathways.
Zhang et al., 2024 [139]	DHAP accumulation under hyperglycemia activates mTORC1/ROS/NLRP3 pathway, inducing podocyte pyroptosis.	This pathway links abnormal glucose metabolism to inflammatory podocyte death, identifying DHAP as a pathogenic factor in DKD.
Zuo et al., 2024 [144]	CCDC92 promotes podocyte lipotoxicity by dysregulating lipid homeostasis via ABCA1 signaling, leading to lipid accumulation and podocyte damage in diabetic kidney disease.	CCDC92 is upregulated in diabetic kidney disease and contributes to lipid deposition and podocyte injury by altering lipid metabolism. Its deletion reduces lipid accumulation and improves podocyte integrity, indicating its potential as a biomarker and therapeutic target.
Yamashiro et al., 2024 [21]	ERK activation in podocytes under high glucose conditions contributes to DN pathogenesis via VEGF and ribosomal biogenesis pathways.	ERK activation was confirmed in DN patient podocytes, suggesting involvement in DN via VEGF signaling and ribosomal regulation.
Esselman et al., 2025 [28]	Podocyte loss and mesangial expansion in glomeruli linked to specific lipid markers detected by MALDI IMS and MxIF.	Using MALDI IMS and MxIF, the study maps lipid markers linked to podocyte and mesangial changes in diabetic glomeruli.
Han et al., 2024 [42]	Podocyte hypoxia from severe microvascular injury promotes extracapillary hypercellularity and loss of podocyte phenotype.	Histological analysis links extracapillary hypercellularity with severe hypoxia-induced podocyte damage in DKD.
Gujarati et al., 2024 [44]	KLF6-induced ApoJ secretion from podocytes activates CaMK1D in proximal tubules, restoring mitochondrial function and protecting against injury.	KLF6 enhances podocyte-proximal tubule communication via ApoJ-CaMK1D axis, preserving renal mitochondrial function.
Lei et al., 2024 [50]	Mesangial expansion, podocyte depletion, and Kimmelstiel-Wilson lesions are key structural changes in DN.	AI-assisted pathology confirms mesangial and podocyte alterations as structural predictors of DN severity.
Hu et al., 2024 [53]	DOT1L/PLCL1 pathway dysregulation.	DOT1L expression is reduced in high-glucose conditions. Its overexpression protects against podocyte injury by upregulating PLCL1, which enhances fatty acid oxidation and reduces lipogenesis, mitigating podocyte damage in DKD.
Lv et al., 2025 [73]	TRAIL/DR5-induced PANoptosis.	TRAIL binds to DR5, triggering apoptosis, pyroptosis, and necroptosis (PANoptosis) in podocytes. Deletion of TRAIL/DR5 reduces kidney injury in DKD models.
Li et al., 2024 [81]	SGLT2 expression increases MAMs, impairing podocyte function; AMPK activation by SGLT2 inhibition restores balance.	SGLT2 inhibitors like empagliflozin reduce MAMs and podocyte injury in diabetic mice via AMPK activation.
Pan et al., 2024 [87]	BTG2 modulates autophagy via mTORC1 inhibition and suppresses EMT, reducing podocyte apoptosis.	BTG2 protects podocytes in DKD by linking autophagy regulation with inflammation pathways shared with periodontitis.
Lv et al., 2024 [92]	PVT1 promotes podocyte injury by modulating TRIM56-mediated AMPKα degradation, leading to mitochondrial dysfunction, mtDNA/mtROS release, and NF-κB-mediated inflammation.	PVT1 upregulation in DKD correlates with disease severity. Its deletion in mice reduces mitochondrial damage and inflammation, highlighting PVT1 as a potential therapeutic target.
Rosenbloom et al., 2024 [98]	Mechanism of vacuolar casts is unclear; hypothesized origin includes degenerated RTECs or podocyturia, with vesicles containing aqueous material.	Vacuolar casts are observed in advanced DN with proteinuria and kidney dysfunction, showing fluid-filled vesicles within a cast matrix on microscopy.
Sunilkumar et al., 2025 [102]	REDD1 reduces slit diaphragm proteins (podocin, nephrin), upregulates TRPC6 and Ca^2+^ influx, disrupting cytoskeleton via NF-κB.	REDD1 deletion preserves podocyte structure in diabetes, reduces albuminuria and glomerular damage, showing therapeutic promise for DN.
Lu et al., 2024 [103]	Rheb1 deficiency causes mitochondrial dysfunction and podocyte senescence through Atp5f1c acetylation, independent of mTORC1.	Rheb1 loss accelerates DKD progression via mitochondrial dysfunction and senescence, representing a novel therapeutic target.
Sun et al., 2025 [116]	AMPK/PGC-1α pathway and mitochondrial protection	Jinlida granules activate AMPK/PGC-1α, improving mitochondrial homeostasis and reducing podocyte apoptosis, offering renoprotection in diabetic mice.
Ward et al., 2025 [120]	Diabetic nephropathy involves vascular damage, mesangial expansion, glomerular scarring, podocyte loss, tubular atrophy, interstitial fibrosis, inflammatory infiltration, and maladaptive repair from fibroblast and macrophage activation.	The nPOD-K cohort includes kidneys from diabetic and non-diabetic donors, preserved for histological analysis to study DKD pathogenesis and progression.
Li et al., 2024 [130]	Podocyte-derived mRNA ratio (podocin:nephrin) reflects qualitative podocyte changes and correlates with fibrosis severity.	Urinary podocin:nephrin mRNA ratio is elevated in DKD and correlates with tubulointerstitial fibrosis, serving as a prognostic marker.
Wang et al., 2024 [134]	miR-193a suppresses WT1, triggering EZH2/β-catenin/NLRP3 pathway activation and inflammasome assembly, leading to inflammation.	Hyperglycemia induces miR-193a, which downregulates WT1 and activates inflammatory pathways, contributing to podocyte damage.
Zhang et al., 2024 [138]	lncRNA EVF-2 upregulation interacts with hnRNPU, promoting podocyte cell cycle re-entry and inflammation in diabetic nephropathy.	EVF-2 contributes to podocyte injury by modulating transcription and splicing, suggesting a novel target for DN therapy.
Zhou et al., 2024 [142]	Hyperglycemia and stress factors reduce α3β1 integrin and alter GBM, causing podocyte detachment and foot process widening.	Structural changes in podocytes correlate with proteinuria severity and DN classification, highlighting their diagnostic value.
Arslan et al., 2025 [24]	miR-342-3p targets SOX6, contributing to podocyte injury, fibrosis, and tubular loss via PI3K/Akt and TGF-β1 pathways.	The study links increased SOX6 expression and decreased miR-342-3p to renal dysfunction, implicating fibrosis-related pathways in DN.
Angeletti et al., 2020 [33]	DAF loss on podocytes leads to complement activation, C3a/C3aR and IL-1β/IL-1R1 signaling, cytoskeletal changes, and reduced nephrin.	DAF deficiency promotes FSGS-like glomerulosclerosis through complement activation and IL-1β-driven inflammation.
Hudkins et al., 2022 [64]	Podocyte loss, increased mesangial matrix, and mesangiolysis.	In a DN mouse model (BTBR ob/ob), podocyte loss and mesangiolysis were mitigated by atrasentan and losartan, which increased podocyte number and reduced mesangial matrix accumulation.
Li et al., 2025 [68]	Foot process effacement and nephrin autoantibodies.	Increased FPW and reduced nephrin expression are observed in DN + MCD. Autoantibodies against nephrin disrupt the slit diaphragm, and hyperglycemia impairs mitochondrial ATP, damaging cytoskeleton.
Li et al., 2025 [69]	RIPK3-mediated inflammation.	RIPK3 induces podocyte injury through NF-κB p65-mediated inflammatory signaling, independent of necroptosis. Its deletion reduces albuminuria and improves glomerular injury.
Hu et al., 2025 [72]	Lipotoxicity and impaired fatty acid oxidation.	DKD induces podocyte lipid accumulation. Dapagliflozin upregulates ERRα and ACOX1, enhancing fatty acid oxidation, reducing lipid toxicity, and restoring podocyte structure.
Li et al., 2025 [91]	Hyperglycemia-induced podocyte detachment, hypertrophy, and effacement compromising the glomerular filtration barrier. SGLT2 inhibitors mitigate these effects by reducing intraglomerular pressure and preserving actin cytoskeleton integrity.	SGLT2i treatment in DKD patients prevented increases in urinary levels of podocyte-specific molecules (podocin, podocalyxin, synaptopodin), indicating a protective effect on podocyte integrity.
Boi et al., 2025 [105]	CKAP4 deficiency disrupts podocyte cytoskeleton, leading to foot process effacement and detachment from basement membrane.	CKAP4 maintains actin and microtubule organization in podocytes. Its reduction in DKD contributes to cytoskeletal disarray and filtration barrier loss.
Wu et al., 2025 [128]	METTL3 induces m6A modification of MDM2, activating Notch signaling, leading to podocyte dedifferentiation and inflammation.	Targeting METTL3 may prevent MDM2-Notch1 mediated podocyte injury and glomerulosclerosis in DKD.
Pan et al., 2018 [9]	SRGAP2a inactivates RhoA/Cdc42 to suppress podocyte motility, maintaining structure and preventing injury under hyperglycemia or TGF-β stimulation.	SRGAP2a is downregulated in diabetic nephropathy. Its overexpression mitigates podocyte injury and proteinuria in diabetic mice.
Xu et al., 2025 [132]	GPR107 deficiency impairs endocytosis of collagen IV and AT1R, increasing membrane bound AT1R, activating AT1R/Ca^2+^ signaling, and promoting GBM thickening.	GPR107 regulates collagen IV balance in podocytes. Its deficiency leads to collagen accumulation and GBM thickening, suggesting therapeutic potential.
Zhu et al., 2025 [140]	CerS6-derived ceramide binds VDAC1, inducing mtDNA leakage and activating cGAS–STING pathway, promoting inflammation.	CerS6 knockout mitigates glomerular injury and inflammation, indicating its role in immune-mediated podocyte damage.
Zhang et al., 2025 [153]	QRXZYQF activates AMPK signaling, modulating ferroptosis by reducing iron overload, oxidative stress, and lipid peroxidation in podocytes.	This traditional Chinese formula protects against DKD by preventing ferroptosis, offering therapeutic benefits via AMPK activation.

3′UTR = 3′ untranslated region. 67LR = 67-kDa Laminin Receptor (RPSA). ABCA1 = ATP-binding cassette transporter A1. ACE-Is = Angiotensin-converting enzyme inhibitor(s). ACOX1 = Acyl-CoA oxidase 1. ADR = Adriamycin (doxorubicin) nephropathy model. AER = Albumin excretion rate. AGEs = Advanced glycation end-product(s). AIF = Apoptosis-inducing factor. Akt = Protein kinase B. ALK1/ALK5 = Activin receptor-like kinase 1 (ACVRL1)/5 (TGFBR1). AMPK/AMPKα = AMP-activated protein kinase (α = subunidade catalítica). APA = Alternative polyadenylation. ApoJ = Clusterin (apolipoprotein J). APOL1 = Apolipoprotein L1. ARBs = Angiotensin II receptor blocker(s). AT1R = Angiotensin II type-1 receptor. Atg4b = Autophagy-related 4B cysteine peptidase. B7-1/CD80 = Costimulatory molecule B7-1 (CD80). BAMBI = BMP and activin membrane-bound inhibitor. BASP1 = Brain acid-soluble protein 1. BDNF = Brain-derived neurotrophic factor. Bcl-2 = B-cell lymphoma 2 (anti-apoptotic protein). BBB = Blood–brain barrier. BIO = 6-bromoindirubin-3′-oxime. BMP-7 = Bone morphogenetic protein-7. BTBR ob/ob = Black and Tan, Brachyury mouse strain with leptin deficiency (ob/ob). C3a/C3aR = Complement component 3a/C3a receptor. CAMP = Cyclic adenosine monophosphate. CCDC92 = Coiled-coil domain-containing protein 92. cGAS–STING = cyclic GMP–AMP synthase/stimulator of interferon genes. CDK5 = Cyclin-dependent kinase 5. CerS6 = Ceramide synthase 6. CG = Collapsing glomerulopathy. ClC-5 (CLCN5) = Chloride channel 5. CTCF = CCCTC-binding factor. CTLA4-Ig = Abatacept (CTLA-4–Ig fusion protein). Cx43 = Connexin-43. DAF/CD55 = Decay-accelerating factor. DAPA = Dapagliflozin. DCTN1 = Dynactin subunit 1 (p150). DGKα = Diacylglycerol kinase-α. DHAP = Dihydroxyacetone phosphate. DOT1L = Disruptor of telomeric silencing 1-like (H3K79 methyltransferase). DR5/TNFRSF10B = Death receptor 5. DR5-Fc = Soluble DR5 decoy receptor (Fc-fusion). Drp1 = Dynamin-related protein-1. DynII1 = Dynein-1 intermediate chain 1 (DYNC1I1). ECM = Extracellular matrix. EGFR = Epidermal growth factor receptor. eGFR = Estimated glomerular filtration rate. eNOS = Endothelial nitric oxide synthase. ENT2 = Equilibrative nucleoside transporter-2 (SLC29A2). ERAD = Endoplasmic reticulum–associated degradation. ERK/pERK = Extracellular signal-regulated kinase/phosphorylated ERK. ERRα = Estrogen-related receptor-α. ESRD = End-stage renal disease. EVF-2 = Embryonic ventral forebrain 2 (lncRNA). FAO = Fatty-acid oxidation. FFA = Free fatty acid(s). FGF1/9 = Fibroblast growth factor 1/9. FOXO1 = Forkhead box O1. FPW = Foot process width. FSP1 = Fibroblast-specific protein 1 (S100A4). G6Pase = Glucose-6-phosphatase. GADD45B = Growth arrest and DNA-damage-inducible protein β. GAK = Cyclin G-associated kinase. GBM = Glomerular basement membrane. GEC(s) = Glomerular endothelial cell(s). GFR = Glomerular filtration rate. GHR = Growth hormone receptor. GLS2 = Glutaminase-2. GLUT4 = Glucose transporter type 4. GM3 = Monosialodihexosylganglioside GM3. GPR107 = G-protein-coupled receptor 107. GPR43 = G-protein–coupled receptor 43 (FFAR2). GSDMD-N = N-terminal gasdermin-D. GSK3/GSK3β = Glycogen synthase kinase-3 (β = isoforma beta). GSNOR = S-nitrosoglutathione reductase. H3K4me3 = Histone H3 lysine-4 trimethylation. HDAC4 = Histone deacetylase 4. HGF = Hepatocyte growth factor. HG = High glucose. hnRNPU = Heterogeneous nuclear ribonucleoprotein U. ID1/ID3 = Inhibitors of DNA binding 1/3. IGF1/2 = Insulin-like growth factor 1/2. IGF2BP2 = Insulin-like growth factor 2 mRNA-binding protein 2. IKKβ = IκB kinase-β. IL-1β/IL-1R1 = Interleukin-1β/IL-1 receptor 1. ILK = Integrin-linked kinase. IMS (MALDI-IMS) = (MALDI) Imaging mass spectrometry. IRS1 = Insulin receptor substrate-1. JAK2 = Janus kinase 2. JAML = Junctional adhesion molecule-like. KLF6 = Krüppel-like factor 6. LINC01619 = Long intergenic non-coding RNA 1619. LRH-1 (NR5A2) = Liver receptor homolog-1. MALDI = Matrix-assisted laser desorption/ionization. MAMs = Mitochondria-associated membranes. MAPK6 = Mitogen-activated protein kinase 6 (ERK3). MCD = Minimal change disease. MDM2 = Mouse double minute 2 homolog (E3 ligase). METTL3/METTL14 = Methyltransferase-like 3/14 (m6A “writers”). miR-xxx = MicroRNA (e.g., miR-27a, miR-99a-5p, miR-770-5p). Mindin = Spondin-2 (SPON2)—“Mindin”. mTOR = Mechanistic target of rapamycin. mTORC1 = mTOR complex 1. MxIF = Multiplex immunofluorescence. NF-κB = Nuclear factor-κB. NIPP1 = Nuclear inhibitor of protein phosphatase-1 (PPP1R8). NLRP3 = NOD-like receptor family pyrin domain-containing 3. NO = Nitric oxide. NOX4 = NADPH oxidase 4. nPOD-K = Network for Pancreatic Organ Donors with Diabetes—Kidney cohort. Orai1 = Calcium release-activated calcium channel protein 1. OXPHOS = Oxidative phosphorylation. P38MAPK = p38 mitogen-activated protein kinase. PA = Palmitic acid. PDK1 = 3-phosphoinositide-dependent protein kinase-1. PEC(s) = Parietal epithelial cell(s). PEPCK = Phosphoenolpyruvate carboxykinase. PGC-1α = PPARγ coactivator-1α. PGNMID = Proliferative glomerulonephritis with monoclonal IgG deposits. PGRN = Progranulin. PI3K/PI3Kα = Phosphoinositide 3-kinase (α = subunidade catalítica p110α). PKC/PLC = Protein kinase C/Phospholipase C. PLCL1 = Phospholipase C-like 1. POLR2B (“Pol2B”) = RNA polymerase II subunit RPB2. PPARγ = Peroxisome proliferator-activated receptor-γ. PTBP1 = Polypyrimidine tract-binding protein 1. PTEN = Phosphatase and tensin homolog. PTGDS = Prostaglandin D2 synthase. PTIP = Pax transactivation-domain–interacting protein. PV-1 = Plasmalemma vesicle-associated protein-1. PVT1 = Plasmacytoma variant translocation 1 (lncRNA). qPCR = Quantitative PCR. QRXZYQF = Fórmula fitoterápica chinesa—nome completo não especificado no resumo; favor confirmar. RAB3B = RAB3B, member of RAS oncogene family. RAAS/RAS = Renin–angiotensin–aldosterone system/renin–angiotensin system. RARRES1 = Retinoic acid receptor responder 1. REDD1 (DDIT4) = Regulated in development and DNA damage responses 1. Rheb1 = Ras homolog enriched in brain-1. RIOK1 = RIO kinase 1. RPS6 = Ribosomal protein S6. ROS = Reactive oxygen species. RTECs = Renal tubular epithelial cells. RYGB = Roux-en-Y gastric bypass. S100A4 = ver FSP1. S6K/p70 S6K = Ribosomal protein S6 kinase beta-1. Sestrin2 = Stress-responsive protein “Sestrin-2”. SGLT2 = Sodium-glucose co-transporter 2. SHP-1 (PTPN6) = Src homology-2 domain phosphatase-1. siRNA = Small interfering RNA. Sirt1/SIRT1 = Sirtuin-1. SOCE = Store-operated calcium entry. SOD2 = Superoxide dismutase 2 (MnSOD). SPD = Synaptopodin (quando “WT1 e SPD” em IHQ). SREBP1 = Sterol regulatory element-binding protein-1. SRGAP2a = SLIT-ROBO Rho GTPase-activating protein 2A. SS-31 = Elamipretide (peptídeo mitocondrial protetor). SUMO2 = Small ubiquitin-like modifier 2. TFEB = Transcription factor EB. TGF-β1 = Transforming growth factor-β1. TGFBR1/TGFBR3 = TGF-β receptor type 1/3. TIMP2/TIMP3 = Tissue inhibitor of metalloproteinases-2/-3. TNFSF10 (TRAIL) = TNF-related apoptosis-inducing ligand. TRIDENT = Transformative Research in Diabetic Nephropathy (coorte/estudo). TRIM56 = Tripartite motif-containing protein 56. TRPC6 = Transient receptor potential cation channel C6. TXNIP = Thioredoxin-interacting protein. UACR = Urine albumin-to-creatinine ratio. UCH-L1 = Ubiquitin C-terminal hydrolase L1. UCP2 = Uncoupling protein-2. VDAC1 = Voltage-dependent anion channel-1. VEGF-A = Vascular endothelial growth factor-A. VPA = Valproic acid. WT1 = Wilms’ tumor 1. β3-integrin = Integrin beta-3 (ITGB3).

The word-cloud synthesis (Figure 5) condenses the mechanistic corpus into a high-salience vocabulary led by “podocyte” (66 mentions) and second-tier tokens such as “podocytes/injury” (20 each), “activation/via” (19 each), “signaling” (18), and a triad of “stress/inflammation/mitochondrial” (13 each). Terms that anchor the filtration barrier, “nephrin,” “foot,” “process,” “detachment,” “effacement,” “slit,” and “cytoskeleton” (≈7 each), signal a dominant barrier-failure theme, consistent with nephrin depletion and increased permeability in human DN, connexin/integrin derangements, and EMT-linked GBM disengagement [3,49,121,135,154]. Tokens denoting mTOR–ER–autophagy stress (“mTORC1/mTOR,” “autophagy,” “degradation,” “TFEB,” “rubicon”) map onto podocyte injury from mTORC1 hyperactivation and autophagy blockade, with rescue signals when lysosomal flux is restored [41,58,59,155].

The prominence of mitochondrial–lipid vocabulary (“oxidative,” “ROS,” “ceramide,” “FAO,” “fission,” “TRPC6,” “calpain”) reflects lipotoxic ROS injury, defective fatty-acid oxidation, and Ca^2+^-coupled mitochondrial fragmentation [78,106,129,156]. Parallel immune lexemes (“inflammasome,” “pyroptosis,” “complement,” “TRAIL/DR5”) mirror CD80/B7-1 induction, C3a/C3aR-mediated mitochondrial dysfunction, NLRP3 activation, and PANoptosis in podocytes [39,55,88,92]. Epigenetic/RNA regulation is also visible (“miR-,” “METTL3/14,” “PVT1,” “LINC-,” “APA”), aligning with m6A-, lncRNA-, and microRNA-driven rewiring of stress pathways [57,61,70,137]. Finally, structural and hemodynamic tokens (“mesangial,” “GBM,” “thickening,” “proteinuria,” “hyperfiltration”) connect the lexical pattern to classic DN histology and microvascular stress [50,107,114].

Taken together, the frequency-weighted lexicon in Figure 5 independently corroborates the five mechanistic pillars summarized in Table 2, slit-diaphragm/adhesion failure; mTOR autophagy ER disequilibrium; mitochondrial lipid stress; immune/complement/inflammasome activation; and epigenetic–transcriptomic reprogramming, while situating them within the broader morphometric context of mesangial expansion, GBM remodeling, and podocyte depletion [75,96].

Figure 6A organizes the eligible studies into a directional cascade. Upstream drivers, chronic hyperglycemia, hemodynamic overload, dyslipidemia/lipotoxicity, AGEs (“metabolic memory”), microvascular stress, and sterile/viral inflammation, converge on signaling hubs, including mTORC1, ER stress/UPR and autophagy/TFEB, TXNIP, EGFR → p70S6K/RPS6, ERK/VEGF, TGF-β/EMT/Notch, ROCK, HDAC4 → calcineurin, complement C3a/C3aR, and NLRP3/pyroptosis (TRAIL–DR5), as well as epigenetic/RNA programs. These hubs impinge on organellar/structural targets, mitochondria (FAO ↓, fission ↑), lysosome/autophagic flux, ER, actin/α-actinin-4 cytoskeleton, slit diaphragm (nephrin/podocin), and adhesion complexes (Cx43, α3β1-integrin), and culminate in tissue-level readouts: foot-process effacement, podocyte loss, proteinuria, mesangial expansion, and GBM thickening/sclerosis. Representative links include mTORC1-driven mislocalization of slit proteins and autophagy suppression [58,59], TXNIP coupling hyperglycemia to oxidative and EMT programs [13], EGFR–Rubicon autophagy blockade [123], ERK/VEGF activation in human DN podocytes [21], TGF-β/Notch-dependent EMT [121], ROCK and HDAC4 → calcineurin pro-apoptotic axes [84,93], complement-driven mitochondrial dysfunction [88], inflammasome/pyroptosis and TRAIL–DR5-mediated PANoptosis [55,92], and epigenetic/RNA control that rewires these hubs [57,70,137]. At the barrier, nephrin loss, connexin-43 heterogeneity, α-actinin-4 reduction, and phase-dependent α3β1-integrin shifts map to detachment/effacement [3,49,135,142,154], with dynein-mediated nephrin degradation and anti-nephrin autoantibodies acting as amplifiers [157]. The cascade’s terminus aligns with morphometric evidence linking podocyte depletion to albuminuria and classic DN histology [50,75,96,107].

Figure 6B collapses the map into five interacting pillars, slit-diaphragm/adhesion, mTOR–autophagy/ER stress, mitochondrial–lipid injury, immune/complement/inflammasome, and epigenetic, transcriptomic control, revealing dense crosstalk. Examples include mTOR-driven barrier disorganization and autophagy loss [58,59]; lipotoxic ceramides and Ca^2+^ entry (TRPC6 or Orai1 → calpain → Drp1) driving mitochondrial fission/apoptosis [117,129,156]; complement C3a/C3aR activation and DAF loss feeding mitochondrial injury, actin remodeling, and nephrin reduction [33,88]; and writer/lncRNA/miRNA programs (METTL3/14, PVT1, miR-27a/193a, LINC01619) that gate Notch/EMT and stress responses and ultimately affect slit/adhesion components [20,57,61,70,92,158]. The network also contextualizes therapeutic reversibility observed across studies: HGF or BTG2 restoring autophagy/lysosomal flux [41,159], SGLT2 inhibition reducing MAMs and activating AMPK [81], and β-hydroxybutyrate or valproate/GM3 correcting oxidative–lipid signals [40,86].

Figure 6C depicts a representative podocyte under high glucose: TXNIP induction, Wnt/β-catenin and miR-27a–PPARγ–FOXO1 axes, and epigenetic modifiers collectively depress protective transcription and diaphragm/cytoskeletal components [13,14,20,160]. In parallel, complement C3a/C3aR signaling and an IL-1β/IL-1R1 loop amplify mitochondrial stress and actin remodeling, and DAF loss disinhibits C3 convertase [33,88]. These signals converge on slit-diaphragm depletion/mistrafficking (nephrin/podocin), adhesion failure (Cx43, α3β1-integrin), and cytoskeletal instability, precipitating effacement, detachment, podocyte loss, and proteinuria, lesions reproduced in models and observed in human biopsies [49,96,121,135,142,154].

Taken together, Figure 6 provides a results-oriented integration linking systemic drivers to intracellular hubs, organelle/barrier failure, and whole-glomerulus pathology. The panels jointly explain why diverse upstream insults can converge on a limited set of podocyte phenotypes and why targeted interventions at different nodes yield structural and functional rescue.

## 3. Discussion

From 7769 records, 130 studies met eligibility and collectively delineate a coherent, staged cascade of diabetic podocytopathy. Despite design-contingent limitations typical of observational work, methodological quality was moderate-to-high, and the synthesis converges on a limited set of mechanistic axes that repeatedly track with structural and clinical phenotypes. Publication activity and model diversity rose sharply after 2019, with an increasing use of mixed human/in vitro/animal designs that balance clinical relevance with mechanistic depth. Together, these features strengthen the inferential link between systemic drivers and podocyte lesions observed across models and human biospecimens.

Our integration organizes the field into five interacting “pillars”, slit diaphragm/adhesion failure; mTOR autophagy ER disequilibrium; mitochondrial lipid stress; immune/complement/inflammasome amplification; and epigenetic transcriptomic reprogramming. Human and experimental data consistently show that nephrin depletion, α-actinin-4 loss, connexin-43 heterogeneity, and phase-dependent α3β1-integrin shifts destabilize the filtration barrier and promote detachment and foot-process effacement, aligning with proteinuria and classic diabetic lesions [3,49,135,142,154].

Podocyte mTORC1 hyperactivation and proteostasis failure mislocalize slit-diaphragm proteins, induce ER-stress/EMT programs, and suppress autophagy; conversely, broad perturbation of mTOR signaling yields hypertrophy and effacement, illustrating a narrow homeostatic window for podocyte proteostasis [58,59]. Mitochondrial lipid injury is driven by palmitate-ROS toxicity, ceramide VDAC1–cGAS STING signaling, impaired FAO, and Ca^2+^ coupled fission via TRPC6/Orai1 calpain CDK5 Drp1, mechanistically linking metabolic stress to apoptosis [78,117,156,161,162].

Immune amplification spans CD80/B7-1 induction, C3a/C3aR-mediated bioenergetic collapse, NLRP3 pyroptosis, and TRAIL/DR5-dependent PANoptosis, with DAF loss removing complement restraint [33,39,55,88,92,150]. Epitranscriptomic and ncRNA programs, METTL3/14, PVT1/EVF-2, miR-27a/193a, and APA gate Notch/EMT and stress signaling and reshape barrier and organelle transcripts, providing a durable molecular substrate for injury [57,70,137,163].

The corpus substantiates upstream drivers, chronic hyperglycemia, hemodynamic overload, dyslipidemia and “metabolic memory,” microvascular stress, and sterile/viral inflammation, funneled through intracellular hubs that damage mitochondria, lysosomes, ER, actin architecture, slit-diaphragm components, and adhesion complexes. TXNIP links hyperglycemia to oxidative stress and EMT/mTOR activation; EGFR signaling increases Rubicon and suppresses autophagy; ERK/VEGF activation is evident in human diabetic podocytes; ROCK and HDAC4 → calcineurin promote cytoskeletal instability and apoptosis; and inflammasome/complement pathways propagate mitochondrial and actin injury [13,21,84,88,92,93,155]. Two clinically salient amplifiers, dynein-dependent nephrin degradation and anti-nephrin autoantibodies, further erode slit integrity and ATP dependent cytoskeletal homeostasis [157,164]. At the organ level, podocyte density declines inversely with albumin excretion and co-localizes with mesangial expansion and GBM remodeling, reinforcing causality between cell-level injury and whole-glomerulus pathology [50,75,96,107].

Experimental data indicate that glucocorticoids can directly preserve podocyte identity and architecture through specific pathways, e.g., dexamethasone KLF15 mediated restoration of differentiation markers and survival, maintenance of miR-30 to restrain Notch1/p53, and recovery of nephrin/synaptopodin with reduced proteinuria [165,166,167]. Angptl4 is glucocorticoid-sensitive, with sialylation conferring protection against proteinuria [168]. Notably, within our corpus and targeted check, we identified no studies that compare steroid-induced hyperglycemia versus normoglycemia on podocyte outcomes in glomerulonephritis under glucocorticoid therapy. Outside a steroid context, metabolic derangements (fasting glucose/insulin, HOMA-IR) correlate with podocyte injury in obesity-related glomerulopathy [169]. This gap merits prospective, glycaemia-stratified studies with podocyte-level endpoints.

Microangiopathy (arteriolosclerosis, hyalinosis) and ischemic remodeling compound podocyte stress and associate with collapsing patterns and adverse outcomes; early hyperfiltration couples to podocyte depletion and GBM thickening, while endothelial–mesangial crosstalk accelerates fibrosis, recapitulating ascending histologic class [50,94,114]. Mesangial-to-podocyte signals suppress ERAD and nephrin phosphorylation, anatomically embedding barrier failure within intra-glomerular signaling loops; diabetic co-culture models confirm HG/MGO-driven transcriptomic deformation and ECM degradation across GEC–podocyte units [34,36].

Conventional markers (albuminuria, eGFR) incompletely capture early podocyte injury; urinary nephrin, podocin, and podocalyxin, and the podocin:nephrin mRNA ratio track progression and tubulointerstitial fibrosis, while spatial metabolomics (MALDI-IMS/MxIF) localizes lipid signatures to podocyte loss and mesangial expansion [28,52,126,130]. Clinicopathologic resources reinforce these links: in TRIDENT, eGFR correlates most strongly with interstitial fibrosis and glomerular epithelial changes, and nPOD-K enables trajectory studies of histologic progression [85,120].

Mechanism-targeted interventions demonstrate structural dividends that validate the five-pillar architecture. HGF and BTG2 restore autophagy/lysosomal flux; DOT1L–PLCL1 improves lipid handling; β-hydroxybutyrate and valproate/GM3 mitigate oxidative–lipid stress and senescence [41,53,86,149,170]. Immune-axis interventions, CTLA4-Ig, C3aR antagonism, inflammasome blockade, DR5/TRAIL inhibition, attenuate complement/pyroptotic injury and rescue barrier components [39,55,88,92]. Systemic metabolic shifts after Roux-en-Y gastric bypass reverse podocyte dedifferentiation and effacement alongside reductions in albuminuria, emphasizing the modifiability of upstream drivers [27]. These convergences argue for “pillar-informed” combinations, e.g., SGLT2i plus an autophagy enhancer, or complement blockade alongside cytoskeletal stabilizers, tested against structural endpoints such as slit-diaphragm density, podocyte density, and EM-level ultrastructure [58,59,81,88].

Across included studies and targeted checks, gliflozins preserved podocyte architecture and reduced proteinuria in diabetic and hyperglycaemic models, with preliminary human correlates. Empagliflozin decreased foot-process width/effacement, increased podocyte number/density, and reduced albuminuria, in association with reactivation of autophagy and attenuation of oxidative stress [64,171]. Dapagliflozin suppressed podocyte epithelial–mesenchymal transition via down-regulation of IGF1R/PI3K and improved nephrin and albuminuria in STZ mice and in a small human DN cohort [172]. Canagliflozin inhibited TXNIP/NLRP3-mediated podocyte pyroptosis with improvements in albuminuria and serum creatinine [173]. Additional signals include restoration of nephrin/podocin and α-Klotho [174], protection of actin cytoskeleton and podocyte density in a nondiabetic proteinuric model [175], reduced podocyte lipotoxicity in Alport syndrome [176], and decreased urinary albumin with a related SGLT2 inhibitor [177].

Mechanistically, benefits converge on autophagy/AMPK activation and relief of mitochondria–ER stress [171,178], increased fatty-acid oxidation via ERRα–ACOX1 [72], EMT suppression [172], and restraint of inflammasome/pyroptosis [173]. Taken together, these data support SGLT2 inhibitors as a structural metabolic backbone for podocyte protection in hyperglycaemia, suitable for combination with pillar-directed modulators (e.g., autophagy, complement/inflammasome, cytoskeleton).

Strengths of this review include its large contemporary corpus, triangulation across human and experimental systems, and explicit mapping from drivers to morphologic readouts. Limitations reflect heterogeneity in models and outcome definitions, incomplete control for confounding in some observational designs, and underrepresentation of certain geographies and in silico approaches. Some promising axes, e.g., endothelial–podocyte metabolic coupling and gut-derived metabolites, remain supported by fewer studies and warrant deeper, prospective interrogation [19,34].

Priority next steps include prospective, mechanistically stratified human studies that co-measure pillar activity (e.g., complement fragments, ceramide species, mitochondrial injury markers, ncRNA and m6A signatures) with standardized morphometrics; trials of rational combinations aligned to individual pillar activation; and deeper dissection of trafficking and autoantibody amplifiers that directly govern slit-diaphragm integrity [157,179,180]. By aligning systemic drivers with intracellular hubs and organellar targets, the field is increasingly positioned to deliver precision interventions that preserve podocyte identity, adhesion, and mitochondrial fitness, thereby interrupting the progression from effacement to proteinuria and renal decline [50,58,59,88,178].

## 4. Materials and Methods

### 4.1. Ethical Aspects

This work was a secondary study and did not violate any current legislation related to ethics in research on humans and experimental models.

### 4.2. Type of Study and Protocol Record

This was a retrospective secondary study through a systematic review. The study was registered in the “Prospero” database under registration number CRD42020205261 and can be accessed at https://www.crd.york.ac.uk/prospero/display_record.php?ID=CRD42020205261 (accessed on 1 August 2025). It was structured according to the recommendations of the tool Preferred Reporting Items for Systematic (PRISMA 2020) [181].

### 4.3. Search Strategy and Eligibility Criteria

We conducted and reported the review in accordance with PRISMA 2020 (study-selection pathway in Figure 1). Four core databases were searched from 1 January 2001 to 31 July 2025: MEDLINE (PubMed), Embase, Latin American and Caribbean Health Sciences Literature (LILACS), and the Cochrane Library (Cochrane Reviews). To broaden coverage, we also performed supplementary searches of gray literature (websites and organizations) and citation chasing (backward screening of reference lists from included studies and forward citation tracking), following principles from the University of Toronto Libraries’ Grey Literature Search Guide. Although the search window spanned 2001–2025, the earliest eligible publication identified was 2002, so included studies cover 2002–2025; counts for 2025 reflect indexing through July.

Search strategies combined controlled vocabulary and free-text terms for podocytes and diabetic kidney disease, adapted to each database. For MEDLINE, we paired Title/Abstract terms for podocytes (e.g., podocyte, podocytes, “glomerular visceral epithelial cells”) with diabetic nephropathy/kidney-disease terms (e.g., diabetic nephropathy, diabetic kidney disease, diabetic glomerulosclerosis, intracapillary glomerulosclerosis). In Embase, Emtree terms were combined with proximity operators (e.g., podocyt* OR phrases such as glomerul* NEAR/3 visceral NEAR/3 epithelial*), intersected with ‘diabetic nephropathy’ or proximity-linked diabetic kidney-disease terms. In the Cochrane Library, we used text-word/proximity formulations (e.g., podocyt* AND (diabetic NEXT nephropath* OR diabet* NEAR/3 kidney NEXT disease* OR DKD). In LILACS, DeCS and keyword combinations analogous to the above were applied. No design or model filters were imposed at the search stage to avoid missing mechanistic studies; importantly, “biopsy” was not enforced as a mandatory search term.

We included original research (human, animal, in vitro, or mixed-modality) that evaluated diabetes mellitus or a hyperglycemic milieu relevant to diabetic kidney disease; assessed podocyte injury at structural/ultrastructural or molecular levels (e.g., foot-process effacement; podocyte number/density; nephrin/podocin/synaptopodin expression; cytoskeletal or organellar injury); and provided a mechanistic context linking exposure to podocyte outcomes. We excluded non-diabetic kidney diseases; studies without podocyte outcomes or without mechanistic information; narrative reviews, editorials, and letters; conference abstracts lacking extractable data; and records for which the full text could not be retrieved.

Study selection, deduplication, and inter-rater agreement. Search results were imported into Rayyan (https://www.rayyan.ai) for automated de-duplication and blinded dual screening of titles/abstracts, followed by full-text assessment. A consolidated log was maintained in Microsoft^®^ Excel for PRISMA accounting and manual verification of residual duplicates. Two independent reviewers (J.S.S., A.G.B.F.) screened titles/abstracts and assessed full texts against pre-specified criteria; disagreements were resolved by discussion, with a third reviewer (W.F.R.) adjudicating when required. Inter-rater agreement for screening decisions was quantified using Cohen’s kappa (κ) computed in BioEst 5.0, with κ values interpreted using conventional thresholds: <0.00 poor, 0.00–0.20 slight, 0.21–0.40 fair, 0.41–0.60 moderate, 0.61–0.80 substantial, and 0.81–1.00 almost perfect agreement. Where applicable, κ was summarized with 95% confidence intervals and two-sided *p*-values.

### 4.4. Evaluation of Study Selection and Methodological Quality

The search covered 1 January 2001 through 31 July 2025; the earliest eligible publication identified was 2002, so included studies span 2002–2025. Counts for 2025 reflect records indexed through July.

Study selection proceeded in two sequential stages: title/abstract screening and full-text review of records passing the first stage. We considered original research across human, animal, in vitro, and mixed-modality designs, provided that studies evaluated diabetes mellitus (or a hyperglycemic milieu relevant to diabetic kidney disease), reported podocyte-level outcomes (e.g., foot-process effacement; podocyte number/density; nephrin/podocin/synaptopodin; cytoskeletal/organellar injury), and offered mechanistic context linking exposure to podocyte injury. We excluded non-diabetic kidney diseases, studies without podocyte outcomes or without mechanistic information, narrative reviews/editorials/letters, conference abstracts lacking extractable data, and records for which the full text could not be retrieved.

Two independent reviewers (J.S.S. and A.G.B.F.) screened titles/abstracts and assessed full texts against pre-specified criteria; disagreements were resolved by discussion, with a third reviewer (W.F.R.) adjudicating when needed.

Methodological quality (risk of bias) was appraised using the Joanna Briggs Institute (JBI) Critical Appraisal Tools matched to study design (human observational, animal experimental, and in vitro/mixed), using the most recent versions available at the time of appraisal (available at https://jbi.global/critical-appraisal-tools; accessed on 2 August 2025). Item-level responses were summarized as absolute/relative frequencies, and overall JBI scores were expressed as percentages to permit cross-study comparison. Any differences in JBI ratings were resolved by consensus among the reviewers. Aggregate quality results are reported in the Results and Table 1.

### 4.5. Data Analysis and Summarization

All extracted variables were first tabulated and checked in Microsoft^®^ Excel. The absolute and relative frequencies for categorical variables and mean, standard deviation, coefficient of variation, and 95% confidence intervals for continuous variables—were computed to summarize study characteristics and methodological quality. Item-level responses from the Joanna Briggs Institute (JBI) tools were summarized as absolute/relative frequencies, and overall JBI scores were expressed as percentages; aggregate summaries report mean, SD, and CV across studies.

Inferential analyses were limited to temporal trend testing and associations between categorical study features. Temporal trends in annual publication counts were assessed using Spearman’s rank correlation (ρ) between year and number of studies and by simple linear regression of studies per year, reporting slope with 95% CI, R^2^, F statistic, and two-sided *p*-values. Associations among study-model categories were evaluated using Pearson’s chi-square test for proportions. Normality of continuous variables was examined with the Shapiro–Wilk test to guide parametric versus non-parametric summaries. A two-sided significance level of 5% (α = 0.05) was adopted.

Trend analyses and the time-series visualization for Figure 2 were performed in GraphPad Prism, version 9.5.1 (GraphPad Software, LLC, Boston, MA, USA). All other analyses and visualizations were scripted in R within RStudio (Posit) 2025.05.0 (Build 496). For data wrangling and plotting we used ggplot2 (v3.5.2), dplyr (v1.1.4), stringr (v1.5.1), forcats (v1.0.0), and scales (v1.3.0). The alluvial diagram (Figure 4B) was generated with ggalluvial (v0.12.5); the continent-by-model heatmap (Figure 4A) used base R heatmap with palettes from RColorBrewer (v1.1-3). The word-cloud and related text features (Figure 5) were produced with tm (v0.7-16), wordcloud (v2.6), and RColorBrewer (v1.1-3); term–document matrices (DocumentTermMatrix) and frequency tables were exported to CSV, and TF–IDF summaries were computed with tidytext (v0.4.3) and visualized with ggplot2 (v3.5.2). Systems maps and schematic figures (Figure 6A) were created with DiagrammeR (v1.0.11) [Graphviz/DOT] and exported via DiagrammeRsvg (v0.1) and rsvg (v2.6.2) to PNG; the five-pillar interaction network (Figure 6B) was rendered with ggraph (v2.2.1) and igraph (v2.1.3). These schematic maps are literature-curated visual syntheses (edge weights used for layout only) and were not used for statistical inference. All figures generated in R were saved with ggsave (ggplot2 v3.5.2), and intermediate tables underlying the heatmap, alluvial diagram, word-cloud, and TF–IDF panels were exported as CSV to support reproducibility.

## 5. Conclusions

Diabetes mellitus (DM) remains a major driver of chronic kidney damage, with podocyte injury representing a pivotal event in the progression of diabetic nephropathy (DN). Our systematic synthesis reveals not only the mechanistic pathways underlying this process, such as redox imbalance, inflammation, apoptosis, autophagy–ER stress dysregulation, and epigenetic reprogramming, but also the temporal and methodological context in which these discoveries have emerged. Publication activity has accelerated sharply since 2019, with over 80% of all studies appearing in the past six years, underscoring the expanding global focus on podocyte biology in diabetes.

This surge has been accompanied by a clear diversification of study designs. While early work was dominated by human biopsy descriptions, more recent years have adopted integrated, multi-model approaches that combine human data, animal experimentation, and in vitro systems, thereby balancing clinical relevance with mechanistic depth. Human tissue studies remain essential for linking lesions to clinical outcomes, whereas in vitro and animal platforms provide controlled environments for dissecting molecular drivers such as mTOR, TXNIP, and NLRP3. Geographic patterns highlight Asia as the leading contributor, with increasing multinational collaborations that enhance generalizability.

Taken together, these findings indicate that the field has transitioned into a mature, multidimensional research space, where diverse methodologies converge on a reproducible cascade of podocyte injury. This evolution strengthens the evidence base for therapeutic strategies that target slit-diaphragm integrity, mitochondrial and metabolic homeostasis, immune and inflammatory pathways, and epigenetic regulators. By situating mechanistic insights within temporal, methodological, and geographical trends, our study provides a comprehensive and timely framework that can inform both experimental designs and translational efforts to mitigate podocyte injury in diabetic nephropathy. We also identify a critical clinical gap: whether steroid-induced hyperglycemia exacerbates podocyte injury in glomerulonephritis remains untested and warrants dedicated investigation.

Building on these insights, the following Future Perspectives outline key priorities to accelerate discovery and translation.

### Future Perspectives

To translate convergent mechanistic insights into clinical benefit, several priorities should guide the field. Standardized, longitudinal human cohorts with harmonized phenotyping (albuminuria, eGFR), biopsy/spatial readouts, and integrated multi-omics are needed to define molecular endotypes and causal trajectories of podocyte injury.

Because our review required diabetes mellitus, studies of glomerulonephritis under glucocorticoid therapy without diabetes, and any differential effects of glucocorticoid-induced hyperglycaemia, were not captured. Prospective GN cohorts that stratify by glycaemic exposure during steroid therapy and track podocyte biomarkers (e.g., urinary nephrin/podocin transcripts, podocyte-derived extracellular vesicles) and, when feasible, ultrastructural endpoints (electron microscopy) are warranted.

Harmonization across model systems, including reporting standards for in vitro, animal, and mixed-modality designs, will improve reproducibility and facilitate cross-study synthesis. Human-relevant experimental platforms (humanized mice, organoids, co-culture microphysiological systems) should be leveraged for causal interrogation of prioritized pathways (e.g., mTOR–autophagy, complement C3a/C3aR, inflammasome, ceramide metabolism, TRPC6/Orai1-mediated Ca^2+^ signaling). Biomarker development should focus on minimally invasive tools (e.g., urinary nephrin/podocin transcripts, podocyte-derived extracellular vesicles) and regulatory-ready surrogate endpoints that track structural repair. Precision-intervention trials can stratify patients by molecular endotype to test pathway-directed therapies and combinations (e.g., SGLT2i backbones with immune-metabolic modulators).

Data sharing and global collaboration, particularly bridging Asian leadership with underrepresented regions, together with transparent code and repositories, will accelerate validation and generalizability. Advanced analytics, including interpretable AI/ML and causal inference frameworks, should integrate temporal, geographical, and methodological heterogeneity. Collectively, these steps can convert the field’s recent expansion into reproducible, patient-centered gains in diabetic kidney disease.

## Figures and Tables

**Figure 2 ijms-26-08990-f002:**
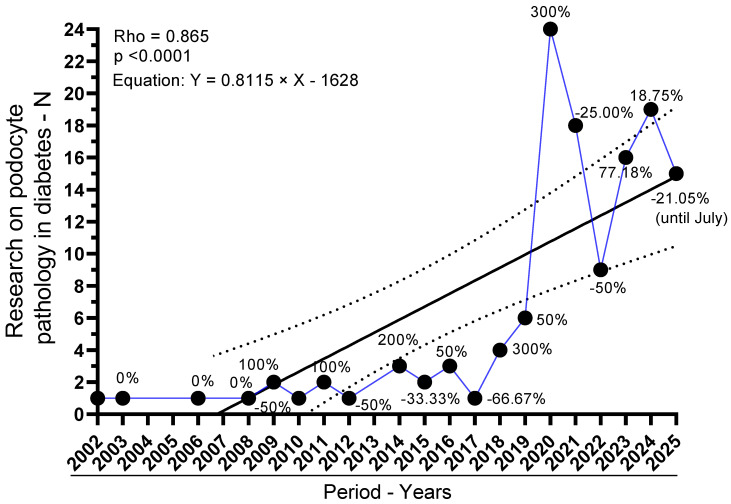
Annual output of studies on podocyte injury in diabetes mellitus (2002–2025). Y-axis: number of studies per year; X-axis: publication year. Black circles mark yearly counts, linked by a blue line for visual continuity. The solid black line is the linear regression fit (Y = 0.8060 × Year − 1617; slope 0.8060, 95% CI 0.4861–1.126; R^2^ = 0.6089; F_1,18_ = 28.02; *p* < 0.0001). Dashed parallel lines depict the 95% confidence bands around the fitted line. A strong monotonic trend is also supported by Spearman’s *p* = 0.87 (95% CI, 0.6814–0.9479; *p* < 0.0001). Counts for 2025 reflect indexing through July only.

**Figure 3 ijms-26-08990-f003:**
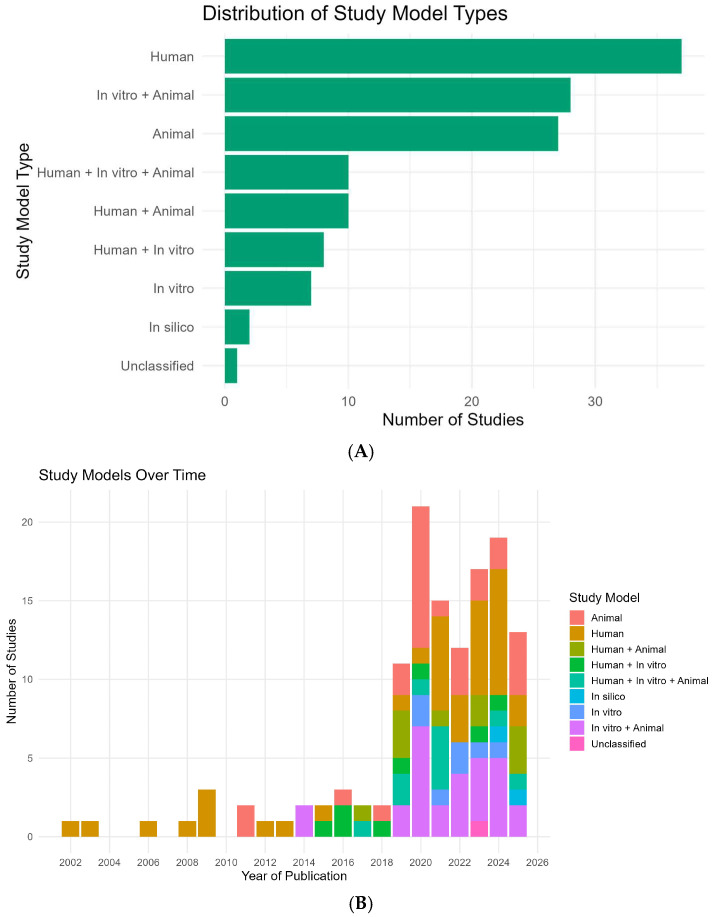
Study models and their temporal distribution (2002–2025). (**A**) Horizontal bar chart summarizing the proportion and count of study models across all eligible records (N = 130). Bars are shown in green; labels at bar ends denote counts. Categories (n, %): Human (37, 28.5%), In vitro + Animal (28, 21.5%), Animal (27, 20.8%), Human + Animal (10, 7.7%), Human + In vitro + Animal (10, 7.7%), Human + In vitro (8, 6.2%), In vitro (7, 5.4%), In silico (2, 1.5%), Unclassified (1, 0.8%). (**B**) Stacked column chart showing, for each publication year, the number of studies stratified by model. The X-axis is the year (2002–2025) and the Y-axis is the number of studies. Colors encode the study models as indicated in the legend (Human, Animal, Human + Animal, Human + In vitro, Human + In vitro + Animal, In silico, In vitro, In vitro + Animal, Unclassified). The 2025 bar reflects records indexed through July only.

**Figure 4 ijms-26-08990-f004:**
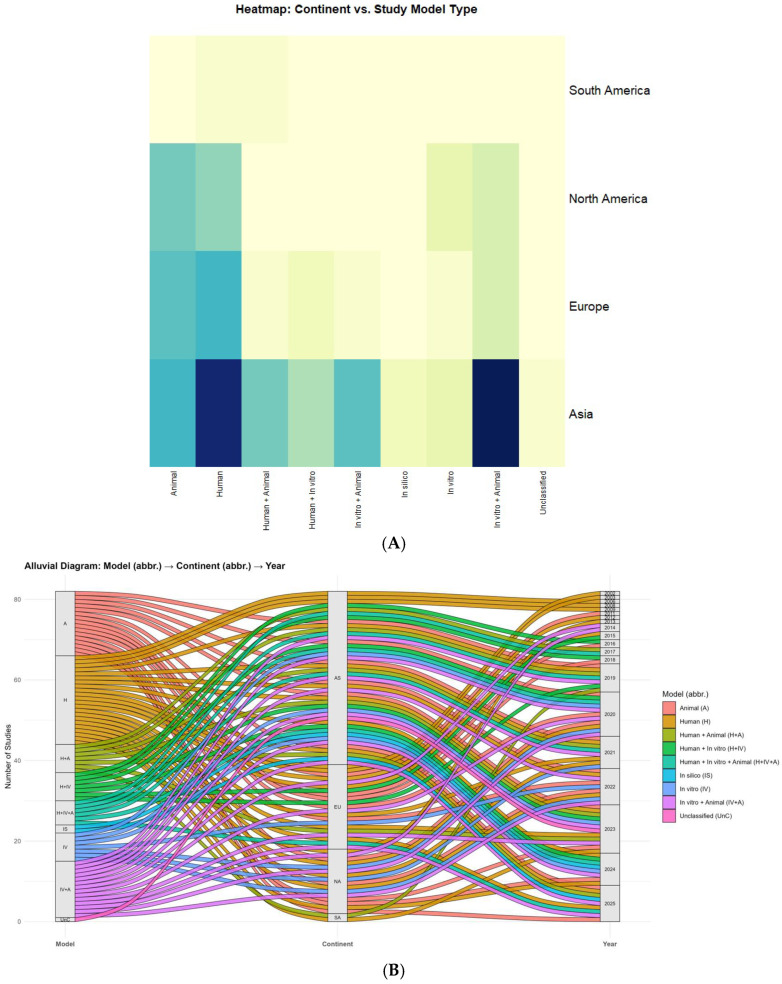
Global distribution of study designs across continents and time. (**A**) Heatmap—Continent × study model. Matrix of absolute frequencies with rows (continents) and columns (model types); darker shades indicate a stronger association (i.e., higher frequency) between continent and model. Clustering is disabled and values are unscaled. (**B**) Alluvial—Model → Continent → Year. Sankey-style flows in which band width is proportional to the number of studies linking model type to continent and publication year; strata are labeled, and color encodes model category. Abbreviations: A = Animal; H = Human; H+A = Human + Animal; H+IV = Human + In vitro; IV = In vitro; IV+A = In vitro + Animal; IS = In silico; UnC = Unclassified; AF = Africa; AS = Asia; EU = Europe; NA = North America; SA = South America; OC = Oceania.

**Figure 5 ijms-26-08990-f005:**
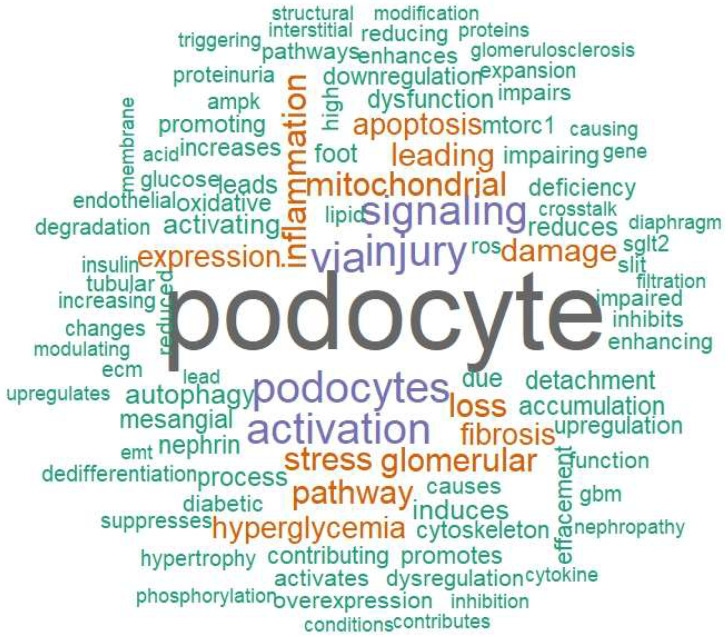
Mechanistic vocabulary of podocyte injury in diabetes. Word cloud summarizing the top 100 single-word terms extracted from the mechanistic sections of all eligible studies. Font size is proportional to term frequency; color is purely esthetic (no quantitative meaning) and word placement is random. Singular/plural or closely related variants appear as separate tokens when present in the source text. The emergent lexicon maps onto the five mechanistic domains summarized in Table 2, slit-diaphragm/adhesion failure, mTOR–autophagy/ER stress, mitochondrial–lipid injury, immune/complement/inflammasome activation, and epigenetic–transcriptomic regulation—providing a compact qualitative overview of the corpus.

**Figure 6 ijms-26-08990-f006:**
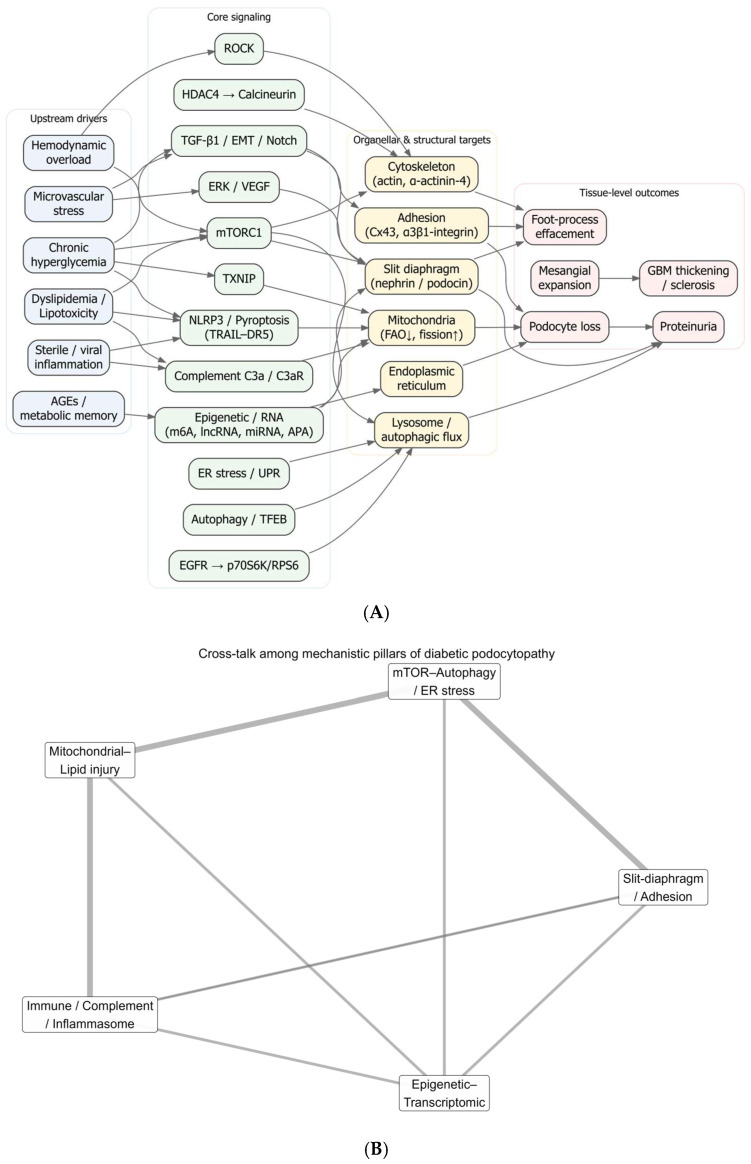
Integrated architecture of diabetic podocytopathy. (**A**) Systems map (drivers → hubs → targets → outcomes). Directed graph summarizing hypothesized causal flow from upstream drivers (light blue: chronic hyperglycemia, hemodynamic overload, dyslipidemia/lipotoxicity, AGEs/metabolic memory, microvascular stress, sterile/viral inflammation) to core signaling hubs (light green: mTORC1, ER stress/UPR, autophagy/TFEB, TXNIP, EGFR → p70S6K/RPS6, ERK/VEGF, TGF-β/EMT/Notch, ROCK, HDAC4 → calcineurin, complement C3a/C3aR, NLRP3/pyroptosis/TRAIL-DR5, epigenetic/RNA programs) and then to organellar/structural targets (light amber: mitochondria—FAO ↓/fission ↑; lysosome/autophagic flux; ER; actin/α-actinin-4 cytoskeleton; slit diaphragm, nephrin/podocin; adhesion—Cx43 and α3β1-integrin), culminating in tissue-level outcomes (light rose: foot-process effacement, podocyte loss, proteinuria, mesangial expansion, GBM thickening/sclerosis). Arrow direction encodes putative influence; edges are drawn only where supported in the corpus (see Table 2). (**B**) Five-pillar interaction network. Circular network collapsing panel A into five interacting modules,) slit-diaphragm/adhesion, mTOR–autophagy/ER stress, mitochondrial lipid injury, immune/complement/inflammasome, epigenetic transcriptomic control. Edge thickness is proportional to the curated crosstalk weight (relative evidence), highlighting dense bidirectional coupling among pillars. (**C**) Cell-level schematic under hyperglycemia. Representative podocyte blueprint depicting hyperglycemia-induced programs (e.g., TXNIP; Wnt/β-catenin and miR-27a–PPARγ–FOXO1 axes; complement C3a/C3aR and IL-1β/IL-1R1 signaling) converging on slit-diaphragm depletion/mistrafficking (nephrin/podocin), adhesion defects (Cx43, α3β1-integrin), cytoskeletal instability, impaired autophagy/lysosome, and mitochondrial dysfunction (FAO ↓, fission ↑), which together yield effacement, detachment, podocyte loss and proteinuria. Abbreviations: AGE, advanced glycation end-product; AMPK, AMP-activated protein kinase; APA, alternative polyadenylation; C3aR, complement C3a receptor; DR5, death receptor 5; EMT, epithelial–mesenchymal transition; ER, endoplasmic reticulum; FAO, fatty-acid oxidation; GBM, glomerular basement membrane; HDAC, histone deacetylase; IL-1R1, interleukin-1 receptor type 1; lncRNA, long non-coding RNA; m6A, N6-methyladenosine; mTORC1, mechanistic target of rapamycin complex 1; NLRP3, NLR family pyrin domain containing 3; TFEB, transcription factor EB; TXNIP, thioredoxin-interacting protein; VEGF, vascular endothelial growth factor.

## Data Availability

The original data presented in this study are fully included in the article. Further inquiries can be directed to the corresponding author.

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
