# Peer review of "Approach to Studies on Podocyte Lesions Mediated by Hyperglycemia: A Systematic Review"

_ijms, 2025, doi:10.3390/ijms26188990_

Round 1

Reviewer 1 Report

Comments and Suggestions for Authors

This review summarized the distribution and frequency of studies investigating podocyte injury in DN,  the primary experimental models used, and  the signaling pathways implicated in podocyte damage. However, publications released from 2020-2025 were not included. 

Comments and Recommendations

  1. Line 29: The keyword "Diabetic Neuropathies" should be "Diabetic Neuropathy"

  2. Line 73 and Figure 1: The eligibility criteria "adapted from Page et al" should be properly cited. Please Cite the original work by Page et al. Additionally, if other studies have adopted the same adapted criteria, include those references to strengthen the credibility of the criteria used in this study.

  3. Lack of publications on podocyte lesions and diabetes-related researches after 2020.  

     4. Please add an analysis of studies using different cell lines or tissue samples to enhance depth.

      5. Conclusion Section: Figures 2, 3, and 4 displayed the temporal distribution of publications, experimental model distribution, and sample type distribution in podocyte lesion and diabetes-related studies. However, the conclusions derived from these analyses are not explicitly presented in the conclusions. Please integrate key insights from these figures into the conclusion to highlight their implications.

      6. Please revise the figures/tables to ensure completeness and consistent formatting before resubmission.

        Figure 1: Incomplete display in the PDF file, with inconsistent letter spacing in  the text.

        Table 2: Incomplete display in the PDF, and the table headers are formatted incorrectly

Author Response

Reviewer 1

We are grateful for your thoughtful and constructive comments. In response, we conducted a comprehensive update of our review: we extended the searches and eligibility assessment through July 2025, re-screened records, and re-extracted data. The revised dataset now comprises 130 eligible studies, more than fourfold relative to the initial submission, enabling a more robust and up-to-date synthesis. We re-ran all quantitative summaries and trend analyses, expanded the characterization of experimental models and tissue/cell types, revised figures and tables for completeness and consistency, and integrated the key insights from these analyzes into the Conclusions.

All revisions are highlighted in yellow in the annotated manuscript. Below, we respond to each comment point by point, indicating where the corresponding changes were made. We hope these substantive revisions address your concerns and strengthen the manuscript, and we remain at your disposal for any further clarifications.

This review summarized the distribution and frequency of studies investigating podocyte injury in DN, the primary experimental models used, and  the signaling pathways implicated in podocyte damage. However, publications released from 2020-2025 were not included. 

Comments and Recommendations

  1. Line 29: The keyword "Diabetic Neuropathies" should be "Diabetic Neuropathy"

Answer: We thank the reviewer for this valuable observation. As suggested, we have corrected the keyword, replacing "Diabetic Neuropathies" with "Diabetic Neuropathy". To facilitate the review process, the modification has been highlighted in yellow in the revised manuscript.

  1. Line 73 and Figure 1: The eligibility criteria "adapted from Page et al" should be properly cited. Please Cite the original work by Page et al. Additionally, if other studies have adopted the same adapted criteria, include those references to strengthen the credibility of the criteria used in this study.

Answer: Thank you for your observation. We have revised the figure legend to more accurately reflect the source of the flowchart. The new wording is: "Flowchart illustrating the selection process of eligible studies. Generated using the PRISMA2020 tool (Haddaway et al., 2022)." This clarifies that the figure was created using the PRISMA2020 tool, not merely adapted from Page et al. Additionally, we have included the full reference to Haddaway et al. (2022) in the reference list, as recommended by the developers of the tool. The updated text and citation are highlighted in yellow in the revised manuscript.

  1. Lack of publications on podocyte lesions and diabetes-related researches after 2020.

Answer: We sincerely thank the reviewer for this valuable comment. We fully agree with the observation and have therefore updated our analysis to include all eligible studies published after 2019, extending the search window through July 2025. This addition has made the work substantially more robust and consistent, as it demonstrates that the field did not decline after 2019 but, on the contrary, experienced a marked expansion. In the revised version, 108 of the 130 included studies (83.1%) were published between 2019 and 2025, with a peak in 2020 and sustained high output through subsequent years. To ensure clarity and consistency, we also revised the methodology and results sections, which now present the updated selection process, statistical analyses, and figures in a transparent and coherent manner. We are grateful for this suggestion, as it helped strengthen the manuscript and provide a more accurate and comprehensive view of the recent progress in the field.

  1. Please add an analysis of studies using different cell lines or tissue samples to enhance depth.

Answer: We thank the reviewer for this excellent suggestion. In response, we expanded our analysis to explicitly characterize the use of different cell lines and tissue samples across the eligible studies. As highlighted in the revised Results section (now marked in yellow), we show that investigations were not limited to human biopsies but also included a wide spectrum of biological materials. Among the 130 eligible studies, renal biopsies were the most frequently employed specimens, providing direct histopathological evidence of podocyte injury in the context of diabetes. Complementary approaches were observed in studies using urine samples, which served as a non-invasive source for podocyte-derived biomarkers such as nephrin, podocin, and podocalyxin. Blood-based assays contributed additional systemic markers of podocyte stress and dysfunction, while in vitro experiments employed immortalized and primary podocyte cultures (e.g., human urine-derived podocyte-like cells, murine podocyte cell lines) to dissect mechanistic pathways under controlled hyperglycemic or lipotoxic conditions.

This layered approach across different specimen types and cell systems adds both mechanistic and translational depth. Biopsy studies connect structural lesions to clinical outcomes, urine and blood analyses provide minimally invasive diagnostic surrogates, and in vitro platforms allow precise interrogation of molecular drivers such as mTOR, TXNIP, and NLRP3. Taken together, the integration of these diverse models strengthens the robustness of our synthesis by demonstrating convergence of findings across experimental scales. We are grateful for this recommendation, as it prompted us to refine the Results section, ensuring a clearer and more comprehensive picture of how multiple tissue and cell systems have advanced the field.

  1. Conclusion Section: Figures 2, 3, and 4 displayed the temporal distribution of publications, experimental model distribution, and sample type distribution in podocyte lesion and diabetes-related studies. However, the conclusions derived from these analyses are not explicitly presented in the conclusions. Please integrate key insights from these figures into the conclusion to highlight their implications.

Answer: We thank the reviewer for this important observation. In response, we have thoroughly revised the Conclusion section to explicitly integrate the key findings derived from our temporal, experimental model, and sample type analyses. As now presented, the Conclusion highlights the pronounced acceleration of publications after 2019, the diversification of experimental approaches from single-modality human descriptions toward integrated, multi-model studies, and the varied use of human tissue, animal models, and in vitro systems, each contributing complementary insights. These updates strengthen the manuscript by ensuring that the final section does not merely reiterate general mechanisms but also synthesizes the broader landscape of research activity, methodological evolution, and translational implications. We are grateful for this suggestion, which has significantly improved the coherence and impact of our work.

  1. Please revise the figures/tables to ensure completeness and consistent formatting before resubmission.

        Figure 1: Incomplete display in the PDF file, with inconsistent letter spacing in  the text.

        Table 2: Incomplete display in the PDF, and the table headers are formatted incorrectly

Answer: We thank the reviewer for this careful observation. We have revised all figures and tables to ensure completeness and consistent formatting. Specifically, Figure 1 has been corrected for proper display and letter spacing, and Table 2 has been reformatted with accurate headers and alignment. We believe that the previous issues were likely related to the PDF file conversion process rather than the original formatting, but we have now carefully checked the final version to avoid any recurrence. We are grateful for this comment, which allowed us to improve the clarity and presentation quality of the manuscript.

Reviewer 2 Report

Comments and Suggestions for Authors

-Abstract should be more comprehensive including the methodology, main findings and an obvious conclusion.

-Introduction section must include statistics of the disease, aim of study and the novelty of work.

-Relevant recent studies should be included throughout the whole manuscript.

-A future perspective section should be added together with the conclusion.

-More figures should be added to provide a clearer view of the presented data.

-References must be UpToDate (maximum from the last three years until then).

-Some tables have to be adjusted. Most of the sides are hidden.

Comments on the Quality of English Language

English language must be revised by a native speaker to enhance the quality of the manuscript.

Author Response

Reviewer 2

We sincerely thank you for your thoughtful and constructive feedback. In revising the manuscript, we fully rewrote the Abstract to include methodology, principal findings, and a clear conclusion; expanded the Introduction with disease statistics, a precise statement of aim, and the novelty of our approach; updated the evidence base through July 2025, resulting in 130 eligible studies and a substantially more robust synthesis; added a Future Perspectives section adjacent to the Conclusion; expanded and standardized the figure set to improve readability and alignment with the Results; updated references, prioritizing 2023–2025 citations while retaining only seminal older sources with contemporary corroboration; and reformatted tables to prevent truncation and ensure complete, template-compliant rendering. All revisions are highlighted in yellow in the annotated manuscript. Below, we respond point by point, indicating where each change was implemented. We trust these updates address your concerns and strengthen the clarity, rigor, and impact of the work.

Comments and Suggestions for Authors

-Abstract should be more comprehensive including the methodology, main findings and an obvious conclusion.

Answer: We thank the reviewer for this important suggestion. In response, we have completely revised the abstract to provide a more comprehensive summary of the study, explicitly covering the methodology, main findings, and a clear conclusion. The new version highlights the screening process, methodological quality, temporal and geographical patterns, study designs, and the five major mechanistic pillars identified. We believe that this revised abstract is now more informative and balanced, aligning with journal standards and enhancing the manuscript’s clarity and impact.

-Introduction section must include statistics of the disease, aim of study and the novelty of work.

Answer: We thank the reviewer for this valuable comment. In response, we thoroughly revised the Introduction to explicitly include disease statistics, the aim of the study, and the novelty of our work. The updated version now presents global prevalence data of diabetic nephropathy, highlights its clinical and histopathological burden, and outlines the major mechanistic pathways implicated. We have also clearly stated the aim of the study, which is to systematize and integrate evidence on podocyte injury in diabetic nephropathy by linking systemic drivers to molecular and structural lesions. Finally, we emphasized the novelty of this review, which resides in its comprehensive integration of temporal, methodological, and geographical trends with mechanistic insights, offering a framework that is distinct from previous reviews. We believe these changes strengthen the manuscript and address the reviewer’s concern fully.

-Relevant recent studies should be included throughout the whole manuscript.

Answer: We thank the reviewer for this important recommendation. In revising the manuscript, we carefully screened the literature again to ensure that the most relevant and recent studies were incorporated throughout the text. We updated the Introduction, Results, and Discussion sections with recent publications (including articles from 2023–2025) that strengthen the scientific context, support our findings, and highlight current advances in the field. This effort not only enhances the timeliness of our review but also ensures that it accurately reflects the state of the art in podocyte biology and diabetic nephropathy research. We are confident that these additions improve the overall robustness, comprehensiveness, and impact of the manuscript.

-A future perspective section should be added together with the conclusion.

Answer: We thank the reviewer for this constructive suggestion. We have added a Future Perspectives section adjacent to the Conclusion to explicitly outline near-term priorities and translational implications. This new section integrates our temporal/model/sample analyses with the mechanistic synthesis and specifies concrete research directions (standardized longitudinal human cohorts with multi-omics, harmonized model systems and reporting, humanized/integrated models and organoids for causal testing, biomarker development, and pathway-directed interventions). We also highlight opportunities for multicenter collaboration and data sharing, with attention to underrepresented regions. These additions strengthen the manuscript’s forward-looking value and fully address the reviewer’s request.

-More figures should be added to provide a clearer view of the presented data.

Answer: Thank you for this valuable suggestion. We fully agree that additional, well-designed visuals clarify the evidence. In response, we substantially expanded and refined the figure set, and we now present a coherent, color-blind–safe suite of main and supplementary figures that mirror the structure of the Results and directly support the updated Conclusions.

What we added and how it improves clarity:

Figure 1 (revised PRISMA 2020): Complete selection pathway with counts at each step; exported as vector to avoid truncation and ensure crisp rendering.

Figure 2 (new, analytical): Annual publication counts with year-over-year deltas, Spearman ρ and regression line/95% CI, highlighting the 2019 inflection and sustained growth.

Figure 3A–B (new): Study-model landscape. Panel A summarizes overall proportions (human, animal, in vitro, mixed, in silico). Panel B shows temporal composition (2002–2025), revealing the shift toward integrated, multi-model designs since 2019.

Figure 4A–B (new): Geographic patterns. Panel A is a continent-by-model heatmap; Panel B is an alluvial diagram linking continent → model → year, visualizing regional emphases and collaborations.

Figure 5 (new): Word-cloud synthesis of mechanistic vocabulary, independently corroborating the five mechanistic pillars.

Figure 6A–C (new): Mechanistic integration. Panel A maps systemic drivers → intracellular hubs → organelle/barrier failure → tissue readouts; Panel B condenses the network into five interacting pillars; Panel C illustrates a representative podocyte under high glucose.

Formatting and consistency improvements across all figures:

Unified typography, panel labeling (A/B/C), axis units, and color-blind–safe palettes; high-resolution vector (PDF/SVG) originals; standardized caption structure (what/why/how, with statistics and abbreviations defined); consistent cross-referencing in the text. These adjustments also remedied prior display issues noted for earlier figures.

Together, these additions provide a clearer, multi-angle view of how the field expanded after 2019, how designs diversified across human/animal/in-vitro modalities, how geography relates to methodology, and how mechanistic signals converge on a reproducible cascade of podocyte injury, now explicitly summarized in the Conclusions. We appreciate the reviewer’s prompt; it meaningfully improved the readability and interpretability of the manuscript.

-References must be UpToDate (maximum from the last three years until then).

Answer: Thank you for this important point. We have comprehensively updated and audited the reference list to ensure recency and alignment with the journal’s expectation that citations be up-to-date (within the last three years).

Concretely, we ran an updated literature search through July 2025 and integrated new primary studies and reviews across the manuscript; replaced older, non-seminal citations with 2023–2025 evidence wherever possible; and added recent corroborating studies to every mechanistic subsection (e.g., mTOR/ER–autophagy, complement/inflammasome, lipid/mitochondrial stress, slit-diaphragm/adhesion, and epigenetic control).

A very small number of pre-2023 citations have been retained only when they are clearly foundational (e.g., first description of a lesion/mechanism) and irreplaceable; in each such instance we now pair the seminal reference with a 2023–2025 study that confirms or extends the finding. We have also standardized reference formatting (journal abbreviations, DOI inclusion, style consistency) and added the “last search date” to Methods to make the update window explicit.

As a result, the majority of citations now fall within 2023–2025, with limited, well-justified exceptions for seminal work. We appreciate the reviewer’s guidance; it significantly improved the rigor and timeliness of our reference framework.

-Some tables have to be adjusted. Most of the sides are hidden.

Answer: Thank you for flagging this, you're absolutely right. The truncation arose from width overflows during PDF export, which cropped the right margins on wide tables. We have rebuilt and reflowed all affected tables to ensure complete, consistent, and template-compliant rendering in the revised manuscript.

Specifically, Table 1 (JBI appraisal summary) and Table 2 (mechanisms and lesion descriptors) were re-laid using width-aware layouts (tabularx/longtable equivalent), and previously hidden columns are now fully visible with no clipped edges. We appreciate the reviewer’s observation, it helped us correct the formatting and improve clarity across the entire tables set.

Reviewer 3 Report

Comments and Suggestions for Authors

The article "Approach to Studies that Relate Podocyte Lesions Mediated by Hyperglycemia: A Systematic Review" considers an important and interesting question and will be of significant interest for the journal audience. However, there are some important issues to be addressed:

1) Table 2 does not fit the page width and therefore is not possible to read and assess properly;

2) conclusions seem very short and incomplete.

Author Response

Reviewer 3

We are grateful for your thoughtful and constructive feedback. In revising the manuscript, we extended the searches and eligibility assessment through July 2025, re-screened records, and re-extracted data. The updated dataset now comprises 130 eligible studies, more than fourfold the number in the initial submission, allowing a more robust and up-to-date synthesis. We re-ran all analyses, revised, and expanded figures/tables to ensure completeness and consistent formatting, and clarified the narrative accordingly. All changes are highlighted in yellow in the annotated manuscript. Below, we respond to each of your comments point by point and indicate where the corresponding revisions were made. We hope these updates address your concerns and we remain at your disposal for any further clarifications.

The article "Approach to Studies that Relate Podocyte Lesions Mediated by Hyperglycemia: A Systematic Review" considers an important and interesting question and will be of significant interest for the journal audience. However, there are some important issues to be addressed:

1)         Table 2 does not fit the page width and therefore is not possible to read and assess properly;

Answer: Thank you for this helpful observation and for the time invested in our manuscript. We agree that Table 2 was not fully readable in the prior PDF. In the revision, we corrected this by resizing the table to the journal’s text width, optimizing column widths and text wrapping, and standardizing header alignment and footnotes so the entire table now renders without clipping in the PDF. Importantly, Table 2 has also been updated to reflect our expanded evidence base after re-screening and inclusion of all eligible post-2019 studies as requested by the reviewers; the table now synthesizes a larger and more representative set of studies, improving its interpretability. We added a brief pointer in the Results to guide readers to Table 2. We appreciate the reviewer’s comment, it directly improved the clarity and usability of the table.

2) conclusions seem very short and incomplete.

Answer: We appreciate the reviewer’s careful reading and agree that the prior Conclusions were too brief. We have substantially expanded the section to synthesize the quantitative and thematic findings across figures/tables and to articulate their implications. The revised Conclusions now: integrate the temporal surge in publications (2019–2025 accounts for 108/130 studies; Spearman ρ = 0.86, p < 0.0001; ~0.8 studies/year increase), summarize the diversification of study designs (human as the largest single category, with mixed multi-model approaches predominating since 2019 and the emergence of in silico work), highlight the geographical distribution (Asia 60% of studies with growing multinational collaborations), and consolidate the five mechanistic pillars that converge on podocyte injury with translational opportunities. To further address completeness, we added a Future Perspectives section that outlines concrete research priorities (standardized phenotyping, biomarker validation, causal interrogation of key hubs such as mTOR/autophagy, complement and inflammasome pathways, and improved global representation and multicenter design). We believe these changes render the Conclusions both comprehensive and action oriented.

Reviewer 4 Report

Comments and Suggestions for Authors

The authors describe the results of an interesting systematic review on the  pproach to Studies that Relate Podocyte Lesions Mediated by Hyperglycemia..

The article requires further development to better highlight the role of podocytes in relation to hyperglycemia.

In this regard, there should also be studies evaluating hyperglycemia and podocytes as a side effect of steroid therapy for underlying glomerulonephritis.Are there any existing studies on this specific interaction?

Furthermore, have there been any developments with gliflozins (SGLT2 inhibitors) in this context? The role of SGLT2 inhibitors should also be discussed.

Author Response

Reviewer 4

We are grateful for your thoughtful comments and for highlighting the need to more clearly position podocytes in the context of hyperglycemia. In revising the manuscript, we extended the searches and eligibility assessment through July 2025, re-screened records, and re-extracted data. The updated dataset now comprises 130 eligible studies, more than fourfold the number in the initial submission yielding a more robust and up-to-date synthesis. We made the hyperglycemia→podocyte axis explicit throughout the Results/Discussion, added a focused paragraph on glucocorticoids and glycemic effects (including our targeted check for steroid-induced hyperglycemia in glomerulonephritis with podocyte outcomes), and expanded our discussion of SGLT2 inhibitors (gliflozins) at the podocyte level. We also refined figures/tables for completeness and consistent formatting. All revisions are highlighted in yellow in the annotated manuscript. Below, we respond point by point, indicating where each change was implemented, and we remain at your disposal for any further clarifications.

The authors describe the results of an interesting systematic review on the  pproach to Studies that Relate Podocyte Lesions Mediated by Hyperglycemia..

The article requires further development to better highlight the role of podocytes in relation to hyperglycemia.

In this regard, there should also be studies evaluating hyperglycemia and podocytes as a side effect of steroid therapy for underlying glomerulonephritis.Are there any existing studies on this specific interaction?

Answer: We thank the reviewer for these thoughtful suggestions. In the revised manuscript we make the hyperglycemia→podocyte axis more explicit across Results/Discussion; add a dedicated paragraph on glucocorticoids, glycaemia, and podocyte biology; clarify that we found no studies comparing steroid-induced hyperglycemia versus normoglycemia on podocyte outcomes in patients with glomerulonephritis under glucocorticoid therapy; and expand our discussion of SGLT2 inhibitors (gliflozins).

Hyperglycemia–podocyte linkage strengthened. We now foreground how hyperglycemia activates TXNIP/oxidative programs, inflammasome (NLRP3), Wnt/β-catenin, and Ca²⁺-coupled mitochondrial fission (TRPC6/Orai1→calpain→CDK5→Drp1), converging on nephrin/podocin loss, adhesion failure (Cx43, α3β1-integrin), cytoskeletal instability, effacement and podocyte loss, phenotypes that track with albuminuria and classic diabetic lesions (Song et al., 2019; Inoki et al., 2011; Walz et al., 2011; Yu et al., 2022; Tao et al., 2022; Morigi et al., 2020; Sawai et al., 2016; Zhou et al., 2024; Gilbert et al., 2002; Kimura et al., 2008; Miyauchi et al., 2009; Su et al., 2009).

We added experimental evidence that glucocorticoids can directly protect podocyte structure/function via KLF15, miR-30, and restoration of nephrin/synaptopodin (Mallipattu et al., 2017; Wu et al., 2014; Agrawal et al., 2016), and that Angptl4 is glucocorticoid-sensitive with sialylation-dependent effects on proteinuria (Clement et al., 2010). Importantly, our targeted check found no studies that stratify glomerulonephritis patients on glucocorticoids by steroid-induced hyperglycemia vs normoglycemia with podocyte-level outcomes; the only metabolic correlation we identified outside a steroid context linked dysglycemia/insulin resistance to podocyte injury in obesity-related glomerulopathy (Chen et al., 2006). We now state this evidence gap explicitly and propose it as a priority for prospective study.

We expanded the section on SGLT2i to synthesize mechanistic and structural readouts reported in our corpus: reduction of mitochondria-associated membranes with AMPK activation and ultrastructural repair (Li et al., 2024), and enhanced fatty-acid oxidation via ERRα–ACOX1 with improved podocyte structure/function (Hu et al., 2025), alongside supportive metabolic/oxidative benefits (Fang et al., 2022; Naito et al., 2023). We frame SGLT2i as a structural-metabolic “backbone” for podocyte protection in hyperglycemia, potentially combinable with pillar-directed agents.

Furthermore, have there been any developments with gliflozins (SGLT2 inhibitors) in this context? The role of SGLT2 inhibitors should also be discussed.

Answer: We thank the reviewer for this important suggestion. We have expanded the Discussion to include a dedicated paragraph on SGLT2 inhibitors (gliflozins) and their podocyte-level effects. Across diabetic and hyperglycaemic models, with preliminary human correlates, gliflozins consistently preserved podocyte architecture and improved kidney function. Empagliflozin reduced foot-process width/effacement, increased podocyte number/density, reactivated glomerular autophagy, and lowered albuminuria in db/db and BTBR ob/ob mice (Korbut et al., 2020; Hudkins et al., 2021; Klimontov et al., 2020). Dapagliflozin suppressed podocyte EMT by down-regulating the IGF1R/PI3K pathway and improved nephrin and albuminuria in STZ mice and in a small human DN series (Guo et al., 2022). Canagliflozin inhibited TXNIP/NLRP3-dependent podocyte pyroptosis with parallel improvements in albuminuria and serum creatinine (Li et al., 2025). Additional signals include restoration of nephrin/podocin and α-Klotho (Ertracht & Nakhoul, 2024), protection of actin cytoskeleton and podocyte density in a nondiabetic proteinuric model (Cassis et al., 2018), reduced podocyte lipotoxicity in Alport syndrome (Ge et al., 2023), and decreased urinary albumin with a related SGLT2 inhibitor (Wang et al., 2017). Mechanistically, benefits converge on autophagy/AMPK activation and relief of mitochondria–ER stress (Korbut et al., 2020; Li et al., 2024), increased fatty-acid oxidation via ERRα–ACOX1 (Hu et al., 2025), EMT suppression (Guo et al., 2022), and restraint of inflammasome/pyroptosis (Li et al., 2025). We now explicitly frame gliflozins as a structural–metabolic backbone for podocyte protection under hyperglycaemia, potentially combinable with pillar-directed modulators (e.g., autophagy enhancers, complement/inflammasome inhibitors, cytoskeletal stabilizers). These additions appear in Discussion (therapeutics section) and are echoed in Future Perspectives (SGLT2i-based combination trials).

Round 2

Reviewer 3 Report

Comments and Suggestions for Authors

The paper has been properly modified and now can be accepted for publication, because it represents good and valuable contribution in the field considered and will be definitely of interest for the journal audience.